# Bis(zinc(II)-dipicolylamine)-functionalized sub-2 µm core-shell microspheres for the analysis of N-phosphoproteome

Yechen Hu[1,2,3,4], Bo Jiang [1,4✉], Yejing Weng[1,2], Zhigang Sui[1], Baofeng Zhao[1], Yuanbo Chen[1,2], Lukuan Liu[1,2], Qiong Wu[1,2], Zhen Liang[1], Lihua Zhang[1✉] & Yukui Zhang[1]

Protein N-phosphorylation plays a critical role in central metabolism and two/multi-component signaling of prokaryotes. However, the current enrichment methods for O-phosphopeptides are not preferred for N-phosphopeptides due to the intrinsic lability of P-N bond under acidic conditions. Therefore, the effective N-phosphoproteome analysis remains challenging. Herein, bis(zinc(II)-dipicolylamine)-functionalized sub-2 µm core-shell silica microspheres (SiO₂@DpaZn) are tailored for rapid and effective N-phosphopeptides enrichment. Due to the coordination of phosphate groups to Zn(II), N-phosphopeptides can be effectively captured under neutral conditions. Moreover, the method is successfully applied to an *E.coli* and HeLa N-phosphoproteome study. These results further broaden the range of methods for the discovery of N-phosphoproteins with significant biological functions.

[1] CAS Key Laboratory of Separation Science for Analytical Chemistry, National Chromatographic R & A Center, Dalian Institute of Chemical Physics, Chinese Academy of Sciences, 116023 Dalian, China. [2] University of Chinese Academy of Sciences, 100049 Beijing, China. [3] School of Pharmacy, Nanjing Medical University, 211166 Nanjing, China. [4] These authors contributed equally: Yechen Hu, Bo Jiang. ✉email: jiangbo@dicp.ac.cn; lihuazhang@dicp.ac.cn

Protein phosphorylation plays important roles in the regulation of signal transduction and gene expression[1,2]. Besides the most recorded O-phosphorylation on serine, threonine, and tyrosine, the phosphorylation of 1-histidine (1-pHis), 3-histidine (3-pHis)[3,4], lysine (pLys)[5,6], and arginine (pArg)[7,8], termed as N-phosphorylation (N-pho, Fig. 1), has been observed for decades but there is a lack of study. Since firstly reported by Boyer[9–11], pHis has been identified to function in two/multicomponent signaling systems in prokaryotes and lower eukaryotes. A bacterial arginine kinase (McsB) phosphorylates arginine residues of the heatshock regulator (CtsR) in the DNA binding domain, to regulate gene transcription[12,13]. The first documented pLys was found in bovine liver, dating back to 1960s[14–16]. However, still little is known about its prevalence and biological roles[17,18].

The main reason that hinders the study of N-pho is derived from the intrinsic lability of P–N bond. Since the phosphorus d-orbital of π-bonding is in a higher energy shell, the overlap between nitrogen lone pair orbital and the phosphoryl π-bond is scarce. Therefore, the P–N bond is not benefited from the stabilization caused by electronic delocalization. In addition, the basic of nitrogen is preserved, which markedly enhances the leaving ability of the amino group once protonated[19,20].

In modern biological approaches, studies were mainly focused on finding target proteins such as kinases and phosphatases. NME1 and NME2 are the only two mammalian protein histidine kinases reported so far[21], and LHPP is discovered as the protein histidine phosphatase and tumor suppressor[22]. Likewise, McsB/YwIE is the well-known arginine kinase/phosphatase pair[23]. However, neither pLys kinases nor phosphatases were found so far, and histone H1 was the only reported pLys protein[24].

To globally elucidate one kind of N-pho, much effort has been made in preparing effective antibodies. Nevertheless, the acid instability and structural flexibility of N-pho makes it difficult to induce immunization by classical methods. Recently, benefited from stable pHis and pArg analogs, several antibodies have been prepared[25,26]. Fuhs et al.[27] generated the first pHis monoclonal antibody, and some pHis candidates related to cell cycle functions were discovered in mammalian cells. Fuhrmann et al.[28] developed two isosteres of pArg and generated the first high-affinity pan-pArg antibody for screening the chaperone proteins, ClpC and GroEL in B. subtilis DywlE strain. Hunter group identified 425 N-pho sites from HeLa lysates using hydroxyapatite and 1, 3-pHis monoclonal antibodies (HAP/pHis mAbs)[29]. However, the structural multiformity introduced by adding phosphate groups to diverse residues makes it rather difficult to develop highly efficient pan-specific antibodies for the global study of N-phosphoproteome. Besides, specific N-pho variants are usually predisposed by certain antibodies and thus can cause biases. For N-phosphoproteome analysis, antibody-independent methods can solve the above problems. However, due to the instability of

P–N bond under acidic conditions, the currently used enrichment methods for O-phosphopeptides are not preferred for N-phosphopeptides, resulting that the landscape of N-phosphoproteome was still covered[30,31]. Inspired by naturally occurring phosphatases, which provide specific Zn(II)-central enzymatic pockets to bind phosphate units of substrates, bis(zinc(II)-dipicolylamine) molecular (DpaZn) is designed for phosphate targets recognition under neutral conditions[32]. Moreover, phos-Tag beads with Zn(II) were used for enrichment of non-canonical phosphorylation peptides[33,34], which further demonstrated the reliability of 2Zn(II). Therefore, DpaZn molecular functionalized materials show great potential for N-phosphopeptides enrichment under neutral conditions.

In this work, DpaZn-functionalized sub-2-μm core–shell silica microspheres (SiO$_2$@DpaZn) are designed for on-tip N-phosphopeptides enrichment under neutral conditions. A total of 27 N-pho sites are identified from Escherichia coli. In addition, SiO$_2$@DpaZn are applied for the analysis of mammalian N-phosphopeptides. In total, 3384 N-pho sites, containing 611 pHis, 1618 pLys, and 1155 pArg, are identified from HeLa cell lysates. We provide reliable technical support for N-pho studies.

## Results

### Design and evaluation of bis(zinc(II)-dipicolylamine) functionalized sub-2-μm core–shell SiO$_2$ microspheres.

Theoretically under neutral conditions, 2Zn(II)-chelated bis (dipicolylamine) (Dpa) groups, terms DpaZn, could bind with phosphorylated target (including both N-phosphorylated and O-phosphorylated targets) and form the stable 1:1 complex[32]. Each Zn(II) ion is coordinated with two pyridyl nitrogen atoms, a tertiary amine, a phenoxy anion, and an oxygen atom from phosphate group (Fig. 2a inset). Herein, nuclear magnetic resonance (NMR) was applied to evaluate whether DpaZn could recognize N-phosphorylated target in neutral solution. As shown in Supplementary Fig. 1a, in $^1$H-NMR, two peaks corresponding to the two protons on imidazole regions of pHis were observed. Upon the addition of DpaZn, such two signals were upfield-shifted by around 0.1 p.p.m. In $^{31}$P-NMR study, the same upfield shift of phosphorus was also observed (Supplementary Fig. 1b), which validated the recognition between DpaZn and the phosphate group.

Moreover, it is noticed that the P–N bond is also heat-labile even under neutral conditions[35] Therefore, rapid enrichment is preferred to improve the recovery of N-phosphopeptides[36]. Herein, sub-2-μm core–shell SiO$_2$ microspheres were designed as the substrate, not only to achieve the specific enrichment within a shorter time, contributed by the fast mass transfer, but also to provide adequate reaction sites for DpaZn immobilization, ensured by the large specific surface area. Accordingly, we could expect that the combination of DpaZn functional groups and

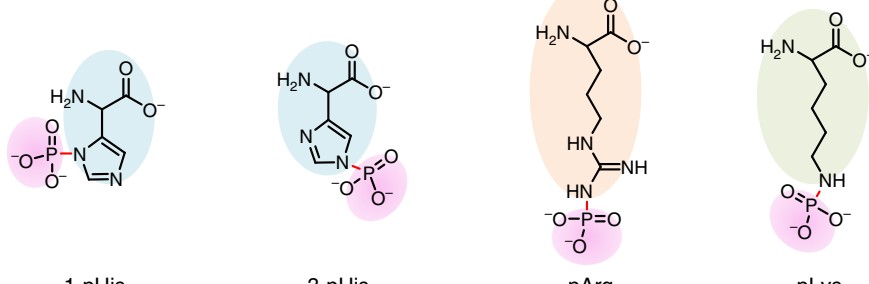

**Fig. 1 Structure of three types of N-phosphorylation.** Phosphorylation occurs at the side chain of peptides histidine, arginine, and lysine, generating P–N bond.

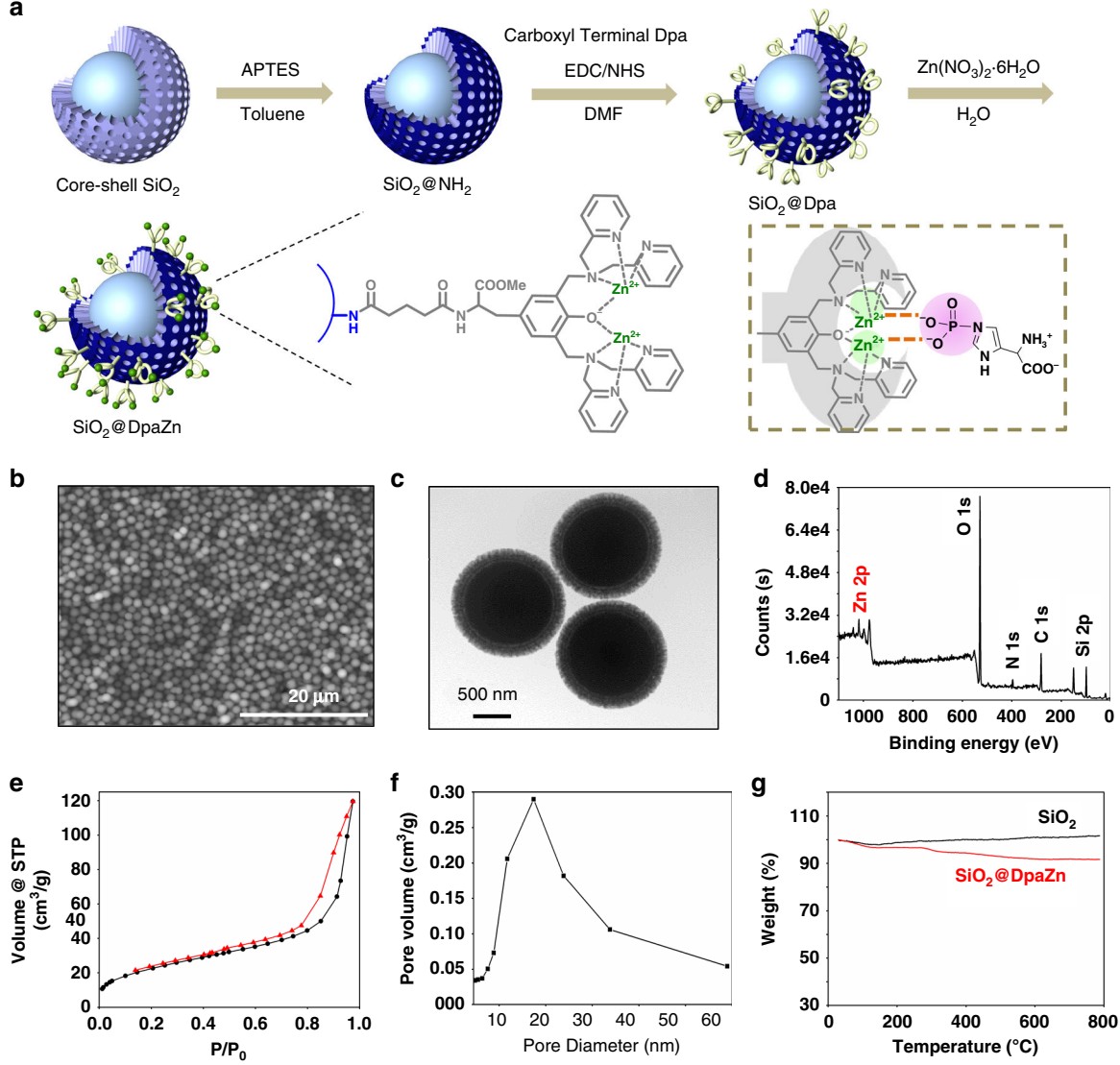

**Fig. 2 Preparation and characterization of SiO₂@DpaZn. a** Schematic illustration (inset shows the interaction between DpaZn and pHis), **b** SEM image, scale 20 μm, **c** TEM image, scale 500 nm, **d** XPS spectrum, **e** adsorption and desorption curves, and **f** pore diameter distribution of SiO₂@DpaZn. **g** TGA curves of SiO₂ microspheres and SiO₂@DpaZn. Three different batches SiO₂@DpaZn were prepared.

core–shell silica microspheres might be effective for the capture of N-phosphoproteome.

SiO₂@DpaZn were prepared by seed growth and template growth methods according to our previous work[37], followed by carboxyl terminal Dpa immobilization and two Zn(II) ions coordination (Fig. 2a and Supplementary Fig. 2). Scanning electron microscopy (SEM) and transmission electron microscopy (TEM) images revealed the highly spherical microspheres with radial core–shell structure, with the core diameter of ∼1.3 μm and the shell thickness of ∼0.16 μm (Fig. 2b, c). The peaks of Zn2p confirmed the existence of Zn(II) ions on the microspheres (Fig. 2d and Supplementary Fig. 3). The surface area and pore size of SiO₂@DpaZn were calculated to be 83.2 m²/g and 17.4 nm, respectively (Fig. 2e, f). The content of DpaZn was calculated as about 14.3% wt (∼143 mg/g) by thermogravimetric analysis (TGA) analysis (Fig. 2g). The above results demonstrated the successful preparation of the material.

The enrichment ability of SiO₂@DpaZn toward N-phosphopeptide was evaluated by capturing pHis peptide TS$_p$HYSIMAR from BSA digests (1:100, m/m) under neutral condition (workflow shown in Supplementary Fig. 4). The peaks of N-phosphopeptide

and its dephosphorylated counterpart could hardly be observed by direct analysis (Fig. 3a). However, after enrichment, most non-phosphopeptides were removed and N-phosphopeptide with enhanced intensity dominated the MS spectra (Fig. 3b), indicating the high enrichment selectivity of SiO₂@DpaZn.

**Establishment of on-tip enrichment strategy**. Commonly in solution enrichment, long incubation time and tedious operation, including vortex and ultrasound, are usually required to ensure the enrichment efficiency. However, the N-pho was rapidly hydrolyzed to 73.2% and 40.1% in these processes within 10 min, respectively (Fig. 3c). Therefore, to improve the recovery of N-phosphopeptides, the entire on-tip enrichment was achieved within 30 min since the adsorption equilibrium by SiO₂@DpaZn could be reached within 5 min (Fig. 3d). By quantitative proteomics (Supplementary Fig. 5), the recovery of N-phosphopeptide by on-tip enrichment was fourfold to that of in-solution enrichment, whereas the total time was shortened to 1/3 (Supplementary Fig. 6), beneficial to achieve the deep-coverage N-phosphoproteome. On-tip method also increased the

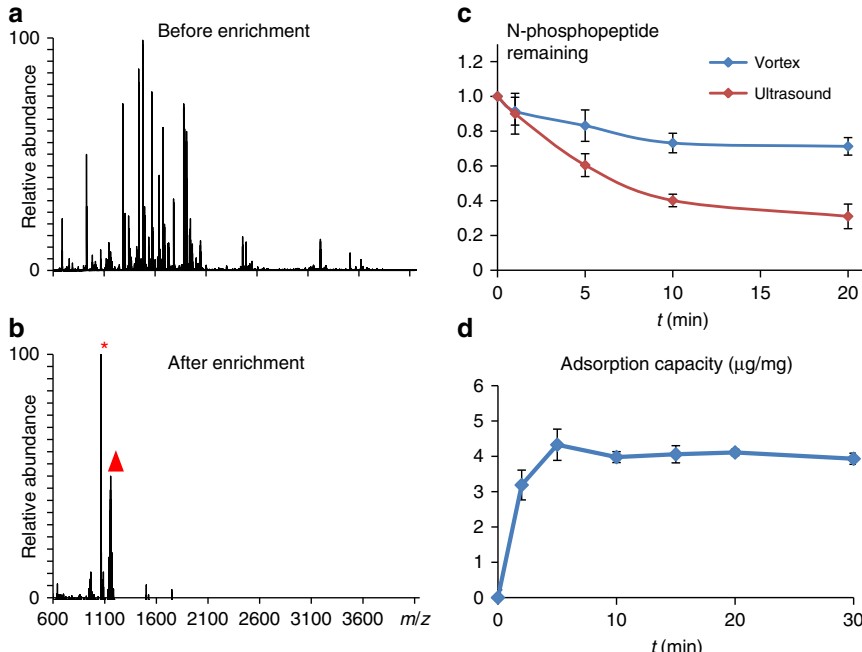

**Fig. 3 Enrichment performance of SiO2@DpaZn. a** MALDI-TOF/TOF spectra of TSpHYSIMAR spiked in BSA digests at 1:100 (m/m) before and **b** after enrichment by SiO2@DpaZn. Triangle indicates the N-phosphopeptide, and asterisk indicates the dephosphorylated counterpart TSHYSIMAR. **c** Degradation curve of N-phosphopeptide in vortex and ultrasound treatment under neutral condition. Each represents the mean from $n = 3$ parallel experiments, and the mean represents recovery of phosphopeptide. The error bar is represented by SD. **d** Adsorption kinetics of SiO2@DpaZn. Each value represents the mean from $n = 3$ parallel experiments, and mean represents adsorption capacity of SiO2@DpaZn. The error bar is represented by SD.

local concentration of DpaZn, which further improved the recognition speed for N-phosphopeptides[38].

**Identification of N-phosphorylated peptides from *E. coli* lysates.** Encouraged by the above results, the comprehensive N-phosphoproteome of Luria–Bertani-cultured *E. coli* was performed after the enrichment by SiO2@DpaZn-based on-tip strategy. Herein, peptide-spectrum matches (PSMs) were strictly checked according to the rules shown in Supplementary Fig. 7. With FDR at peptide and PSMs level set <1%, and the cut-off value of ion score as 20, combined with manual site localization, the direct evidence of 27 N-pho sites (15 pHis, 8 pLys, and 4 pArg) and 12 O-pho sites (Supplementary Dataset 1) was identified after removing the pLys/pArg peptides localized to the C-terminus. Furthermore, 3 pHis sites (PPS H421, manX H20, and ptsI H189) were in accordance with the previous method[39]. To further explore the influence of different carbon sources on N-pho sites in *E. coli*, N-phosphopeptides from M9 minimal medium-cultured *E. coli* (glucose or glycerol) were enriched by our method. After removing the pLys/pArg peptides localized to the C-terminus, 99 N-pho sites (19 pHis, 38 pLys, and 42 pArg) and 101 O-pho sites (39 pSer, 56 pThr, and 6 pTyr) with localization probability over 0.75 were identified (Supplementary Dataset 2). These results demonstrated the reliability of our method to N-phosphoproteome analysis.

To further confirm the authenticity of the N-pho sites identified in our study, seven of the N-phosphopeptides were synthesized (Supplementary Dataset 1). The MS/MS spectrum and retention time of the synthetic peptides were compared with that obtained from in vivo peptides. As an example, autonomous glycyl radical cofactor (YfiD) is documented as an acid-inducible protein in *E. coli* in response to environmental stress[40], and has been reported to be a phosphoprotein with the phosphoresidue(s) unstable to acid[41]. However, the exact modified sites were still unknown. In this study, two N-pho sites (H26 and K48) on YfiD were identified (AEAGIVISASpHNPFYDNGIK and

AGYAEDEVVAVSpKLGDIEYR). The MS/MS spectra of the peptides from in vivo (AEAGIVISAS+79.98DaHNPFYDNGIK and AGYAEDEVVAVS+79.98 DaKLGDIEYR) with a mass shift of +79.98 Da at the histidine and lysine residues had the same MS/MS spectra as that of the synthetic peptides with a phosphate group on histidine and lysine, respectively (AEAGIVISASpHNPFYDNGIK and AGYAEDEVVAVSpKLGDIEYR) (Fig. 4a, b). In addition, the two synthetic N-pho peptides eluted at the same time with the in vivo peptides on HPLC (Fig. 5a, b), confirming that the detected mass shift of +79.98 Da in the in vivo-derived peptides was caused by N-pho. Similarly, the MS/MS spectra and retention time of other N-phosphopeptides were also examined and exhibited in Supplementary Fig. 8. Taken together, we confirmed the authenticity of these reported N-phosphopeptides. Moreover, after label-free quantification by the MaxQuant software, calculated Pearson correlation coefficients (PCCs) for the quantified peptides in triplicate (each replicate corresponding to a different enrichment and different LC-MS/MS injection) were all around 0.90 ($n = 3$, Supplementary Fig. 9), demonstrating our enrichment method is robust and reproducible.

**Identification of N-phosphorylated peptides from HeLa lysates.** To further elucidate the N-pho state of mammalian cells, SiO2@DpaZn was applied for HeLa cell lysates. Totally, 3384 N-pho sites, containing 611 pHis, 1618 pLys, and 1155 pArg, and 6635 O-pho sites, containing 4105 pSer, 1956 pThr, and 574 pTyr, were identified from HeLa lysates (Supplementary Dataset 3), among which 2 pHis sites, including MCM3 3-pHis 721 and LMNB1 1-pHis/3-pHis 571, were verified by virtue of monoclonal pHis antibodies[27]. However, we did not identify any known pHis sites including NME1/2 1-pHis 118, PGAM 3-pHis11, and NDK1 1-pH117 from biological samples. Therefore, to be cautious, we stated that we did not know whether our method enriches both forms of pHis from complex biological samples. Motif analysis is conducive to evaluate the features of the

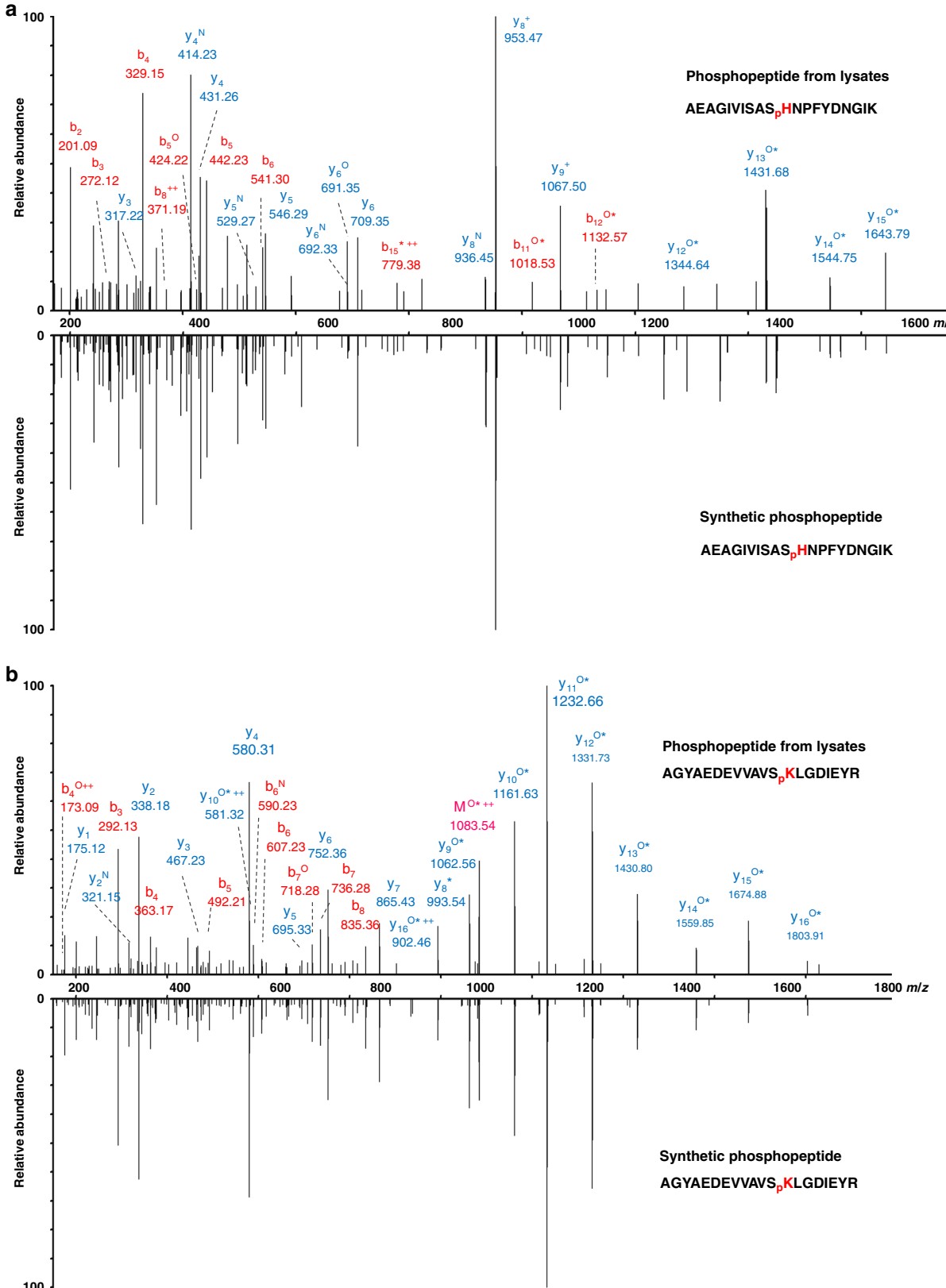

**Fig. 4 MS/MS validation of YfiD pHis and pLys peptides in *E. coli*. a** MS/MS spectra of the endogenous *E. coli* YfiD pHis peptide (AEAGIVISAS$_p$HNPFYDNGIK) and **b** pLys peptide (AGYAEDEVVAVS$_p$KLGDIEYR). The synthetic peptides are shown in the mirror image. The superscript asterisks, N, and O indicate the dephosphorylated (−79.97 Da), deamination (−17.03 Da), and dehydrated (−18.01 Da) ions, respectively. These graphs are made with origin 8.

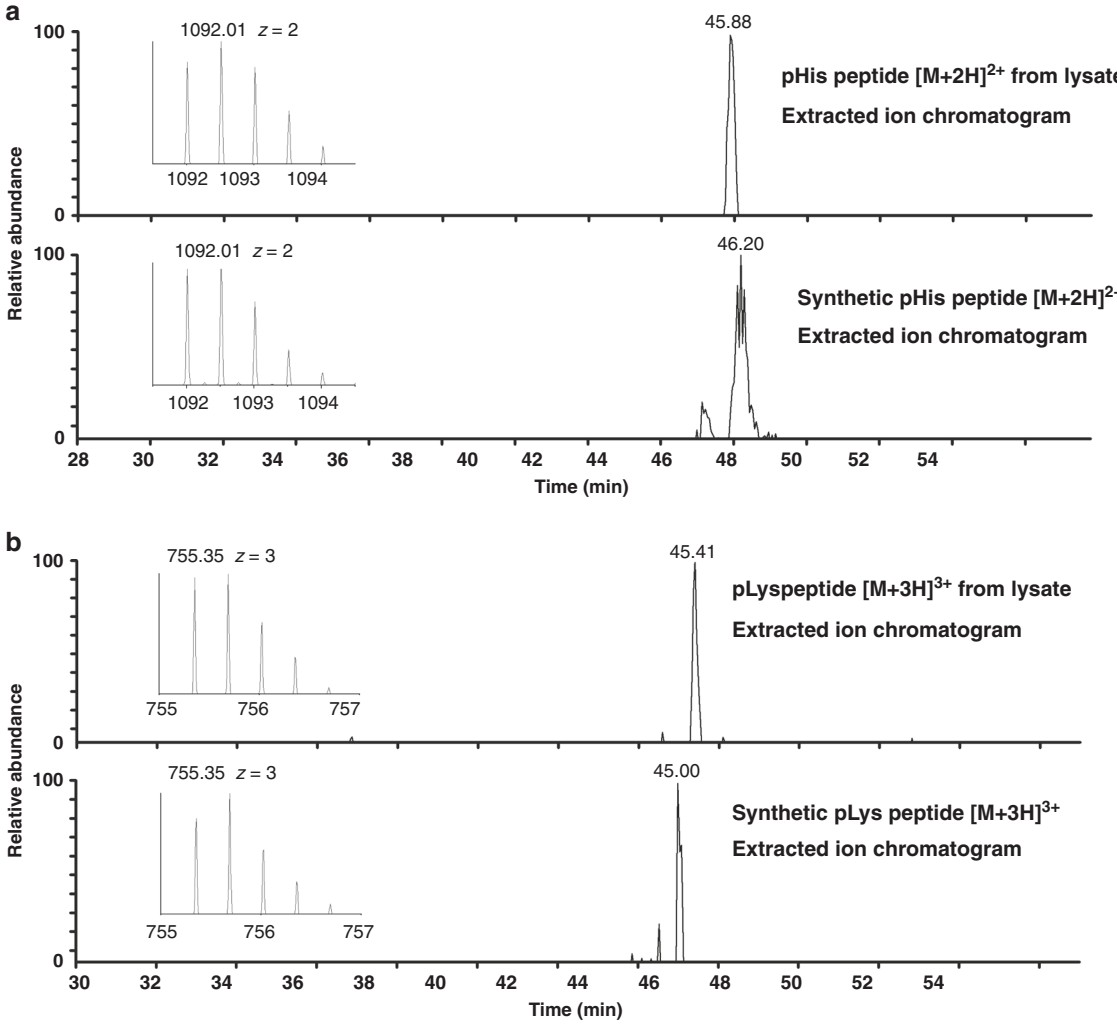

**Fig. 5 HPLC validation of YfiD pHis and pLys peptides in *E. coli*.** Extracted ion chromatogram of **a** the pHis peptide and **b** pLys peptide from *E. coli* lysates (top) and synthetic peptide (bottom), respectively. Insets are the MS spectra of the precursor species at *m/z* 1092.01 and 755.35. These graphs are made with origin 8.

N-phosphoproteins in HeLa cells, and the ±10 residue sequence windows were generated from all N-pho sites and tested against HeLa proteome background. Consistent with the previous studies of pHis, the data exhibited enrichment of leucine residue around the N-pho sites (Supplementary Fig. 10a). This could either be a preference for the His kinase, or alternatively this could be because peptides with leucine residues in the vicinity of the N-pho residue could be more resistant to hydrolysis. The result was further verified in HEPG2 cells (Supplementary Fig. 10b). To elucidate the biological function of N-phosphoproteins identified from HeLa cells, gene ontology (GO) analysis of the identified 2596 N-phosphoproteins was performed, which revealed the significant enrichment of biological process terms of metabolism, regulation of metabolism, organization, and immune response (Supplementary Fig. 11). Besides, we also found that these proteins participated in ATP binding, ATPase activity, RNA binding, nucleotide binding, and protein kinase binding, which was consistent with previous results for HeLa cells[29,42].

**Investigation of the interaction between SiO₂@DpaZn and N-pho peptides**. We made further effort to elucidate the interaction between SiO₂@DpaZn and N-pho under neutral conditions. As shown in Fig. 6a, the enrichment selectivity of TS$_p$HYSIMAR by various materials is in the order of SiO₂@DpaZn > SiO₂@Dpa >

SiO₂@NH₂, indicating that metal ion chelation makes the main contribution to the selectivity, and further enhanced by hydrophilic interaction (Supplementary Fig. 12). We made further confirmation of such deduction by adjusting the mixed ratio of ACN and NH₃·H₂O respectively mixed with the N-phosphopetide, since ACN facilitated the hydrophilic interaction, whereas NH₃·H₂O inhibited the chelation between Zn(II) and phosphate groups. As shown in Fig. 6b, with the decrease of ACN and the increase of NH₃·H₂O, non-phosphopeptides, mono-phosphopeptides (*m/z* 1145, 2065, and 2556,) and the multi-phosphopeptide (m/z 3122) were eluted in sequence, in accordance with our expectation. Moreover, we found the zeta potential of SiO₂@DpaZn kept stable at about +40 mV around pH 7.0 (Fig. 6c), enabling the additional electrostatic interaction with phosphate groups. Collectively, in the recognition process of phosphate groups, the primary coordination bonding with DpaZn was strengthened by additional secondary noncovalent interactions including hydrophilic and electrostatic interactions. To further quantify the interaction between SiO₂@DpaZn and N-phosphopeptides, several in vivo-derived N-phosphopeptides from *E. coli* with different net charges were synthesized and applied to isothermal titration calorimetry (ITC) experiments. The interaction between DpaZn and N-phosphopeptides was endothermic, indicating it an entropy-driven process (Supplementary Fig. 13). Besides, the dissociation constant between DpaZn with four

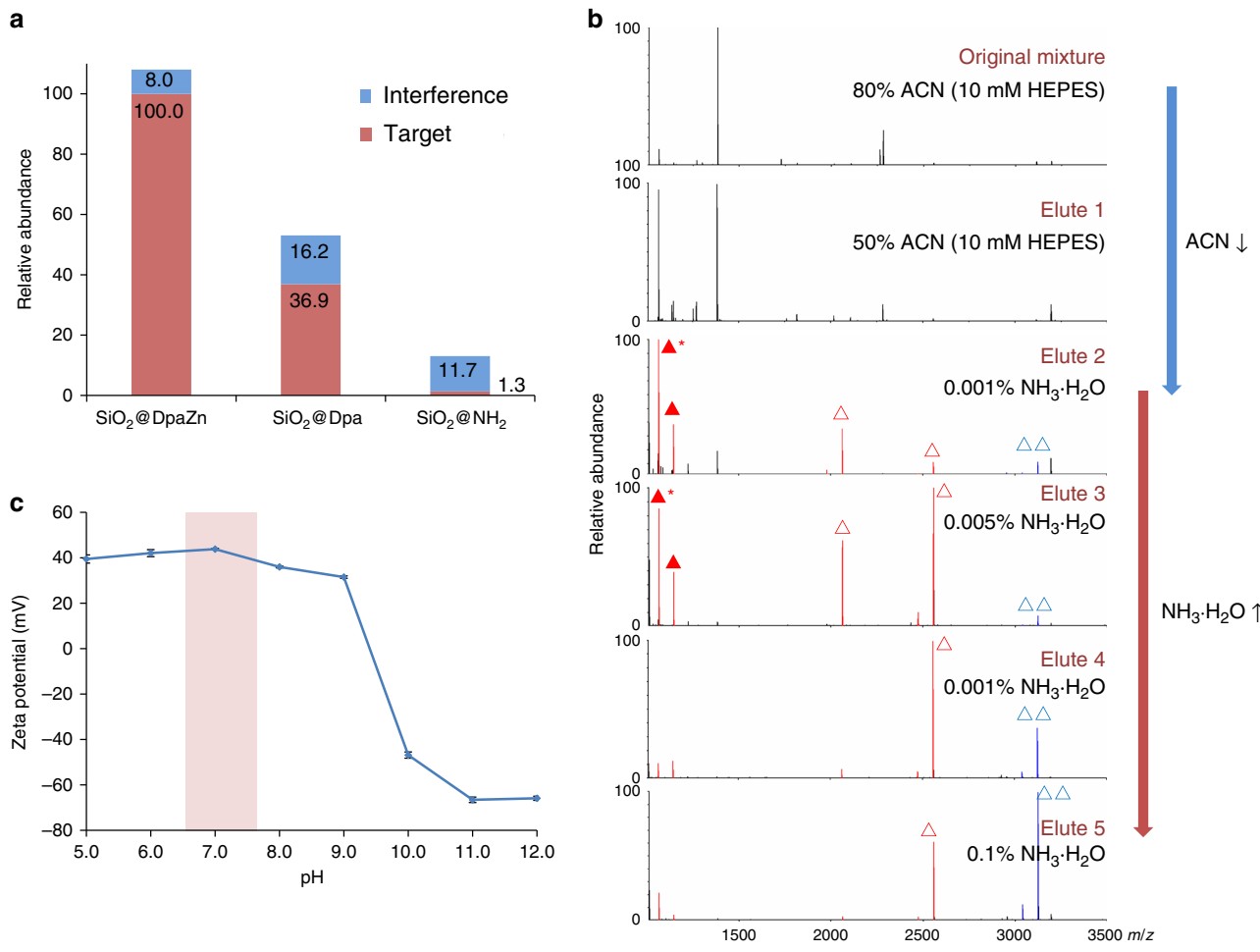

**Fig. 6 Investigation on interaction between SiO$_2$@DpaZn with phosphopeptides. a** Enrichment ability of SiO$_2$@DpaZn, SiO$_2$@Dpa, and SiO$_2$@NH$_2$. **b** Gradient elution of nonphosphopeptides, mono-phosphopeptides (triangle), and multi-phosphopeptides (double triangles). Solid triangle indicates N-phosphopeptide (TS$_p$HYSIMAR), and asterisk-labeled solid triangle indicates the dephosphorylated counterpart (TSHYSIMAR). **c** Zeta potential of SiO$_2$@DpaZn in pH 5.0–12.0. Each value represents the mean value from $n = 3$ parallel experiments. The error bar is represented by SD.

**Table 1 Amino acid sequences of the N-phosphopeptides with different net charges and the stoichiometry ($n$), enthalpy ($\Delta H$, kcal/mol), entropy ($\Delta S$, kcal/mol), and dissociation constant ($K_D$, μM) for the interactions with DpaZn.**

|  | Sequence | Net charge | Gene | $n$ | $\Delta H$, kcal mol$^{-1}$ | $\Delta S$, kcal mol$^{-1}$ | $K_D$, μM |
|---|---|---|---|---|---|---|---|
| Peptide-a | AGYAEDEVVAV S$_p$KLGDIEYP | −5 | grcA | 0.846 ± 0.090 | 1940.0 ± 284.7 | 28.8 | 13.14 |
| Peptide-b | A$_p$KLESLVEDLV NR | −3 | dnaK | 1.010 ± 0.037 | 1719.0 ± 85.3 | 26.2 | 34.01 |
| Peptide-c | QWVNLPLVL$_p$H GASGLSTK | 0 | gatY | 0.844 ± 0.284 | 1671.0 ± 630.2 | 24.1 | 90.91 |
| Peptide-d | Q$_p$KLHGYLPSR | 1 | aceE | 0.899 ± 0.056 | 1284.0 ± 94.6 | 22.4 | 113.64 |

N-phosphopeptides ranged from 13.14 μM to 113.64 μM (Table 1). In the ITC experiments, DpaZn was uniformly distributed in the solution, which was actually inconsistent with the actual on-tip enrichment. Just like the immobilized enzymatic reactor can condense the concentration of enzyme and accelerate digestion[38], SiO$_2$@DpaZn was tightly packed into a micropipette tip to achieve a volume of about 3.8 μL for cell lysates enrichment. Then the actual concentrate of DpaZn in the enrichment system could be calculated as 130.9 mM. From the perspective of kinetics, high concentration DpaZn is conducive for the rapid recognition of low-abundance N-pho peptides.

## Discussion

In this study, SiO$_2$@DpaZn was carefully designed for on-tip enrichment of N-phosphopeptides. In fact, from the identification

results of *E. coli* and HeLa, O-pho peptides were inevitably enriched at the same time. Indeed, SiO$_2$@DpaZn can bind both N-pho peptides and O-pho peptides. However, according to previous papers[43,44], the DpaZn coordination complex has a vacancy on each Zn(II) ion that the phosphate anion [RO(N)PO$_3$] can access to form RO(N)PO$_3$-DpaZn complex. Compared with P–O bond, the N atom of P–N bond has a stronger electron donating ability than the O atom, which makes the structure of RNPO$_3$-DpaZn more stable. Therefore, this might result in N-pho peptides being preferentially enriched over O-pho peptides under the neutral conditions. This also explained the relatively high proportion of N-pho peptides in our results.

Due to the important biological function of N-pho, several groups are committed to develop antibody-independent methods to enrich N-pho peptides. Potel et al. reported a Fe$^{3+}$-IMAC-

**Table 2 The number of N-pho identified from *E. coli* and HeLa lysates by our and other groups.**

| Sample | *E. coli* | | HeLa | | |
|---|---|---|---|---|---|
| Method | SiO$_2$@DpaZn | Fe$^{3+}$-IMAC | SiO$_2$@DpaZn | SAX | HAP/pHis mAbs |
| pHis | 19 | 135 | 611 | 225 | 77 |
| pLys | 38 | – | 1618 | 278 | 174 |
| pArg | 42 | – | 1155 | 278 | 174 |
| Ref. | This work | 45 | This work | 40 | 29 |

Sites localization probability over 0.75 is applied for all data.

based strategy for the analysis of pHis proteome in *E. coli* cultured with glucose and glycerol[45]. Combining mild acidic (pH 2.3) enrichment and pollutant removal strategy including protein precipitation and benzonase treatment, 135 pHis sites were identified (Table 2). Compared with our results, no identical pHis site was found, which might be attributed to different enrichment conditions. However, considering that pLys and pArg suffered from severe hydrolysis under mild acidic conditions or long-term operation[46,47], our method might be more suitable for comprehensive N-pho sites discovery. Furthermore, 27 unphosphorylated peptides in our results were identified as pHis peptides by Potel et al., which might be attributed to acidic separation. Although acidic separation contributes to good separation, it might also increase the risk of N-pho hydrolysis. To overcome the problem, photonic crystals with ultra-efficiency separation might be an alternative to significantly reduce separation time[48].

In additional, a strong anion exchange-based method (SAX) was developed by Hardman et al. to reveal human non-canonical phosphorylation. A total of 781 unique N-pho sites, including 225 pHis, 278 pLys, and 278 pArg, were identified from HeLa lysates (Table 2). Comparing HeLa N-pho results between the existing three methods, low overlap of N-pho sites was found (Supplementary Fig. 14), which might be due to different enrichment mechanisms under neutral conditions and different separation and MS conditions. However, these methods had meaningful complementarities which were beneficial for N-pho sites identification. For example, an interesting result was found about the pArg sites on protein SRRM2. Four pArg sites (R294, R320, R986, and R2103) were found by both SiO$_2$@DpaZn and SAX. Moreover, 3 (R1494, R2131, and R2396), 6 (R302, R356, R851, R1530, R2119, and R2286) and 1 (R1879) pArg sites were exclusively identified by SAX, SiO$_2$@DpaZn, and HAP/pHis mAbs, respectively.

pHis is physiologically important for both bacteria and mammals[49–52]. After re-examining our data, some vital N-pho proteins that perform protein kinases or phosphatases were identified, such as Phosphoenolpyruvate synthase and Phosphoenolpyruvate-protein phosphotransferase. Although several known pHis sites were successfully identified in our study, we failed to find some known p-His sites, including NME1/2 1-pHis118, NDK1 1-pHis 117, and PGAM 3-pHis11. The similar phenomenon was found in a previous report[42,45]. It deserved in-depth analysis, and there were three possible reasons for explaining failure to identification of known pHis sites: (i) The enrichment of N-pho is carried out under neutral conditions, and so negative charge peptides would be co-eluted with N-pho peptides in this case, which might inhibit N-pho peptides identification. (ii) Some tryptic pHis-containing peptides like the histone H4 (H18) and ACLY (H760) peptides are too short or too long to be considered by MS using current parameters[29]. (iii) Acid separation conditions would increase the risk of pHis hydrolysis[29].

In summary, the above results reveal four appealing features of SiO$_2$@DpaZn: specific complexation with N-pho peptides driven by coordinate interaction at moderate binding ability, rapid adsorption within 10 min enabled by core–shell structure, high recovery and modification preservation achieved by on-tip analysis, as well as excellent enrichment ability toward diverse N-pho peptides (especially rare pLys and pArg peptides) from biological samples. A large amount of unique N-pho sites have been identified from *E. coli* and HeLa lysates benefiting from these different characteristics from traditional TiO$_2$ or IMAC materials. High discovery rates for N-pho sites and non-discrimination of recognizing all kinds of phosphorylated peptides illustrate the great potential of our material in comprehensive N-phosphoproteome analysis. Therefore, this universal method might greatly promote the studies of the world of N-phosphoproteins. In addition, the smart core–shell structure design and fast on-tip enrichment concept can be extended to other protein PTMs such as polyphosphorylation[53] and S-sulphenylation[54], which play vital roles in biological processes but are all fragile under certain conditions and cannot be efficiently captured by artificial materials so far. By precise designing the recognition groups, controlling the enrichment conditions, together with applying the fast on-tip enrichment, these challenges might be addressed. Furthermore, if the enrichment pH value was reduced to 2.3, SiO$_2$@DpaZn may be suitable for large-scale pHis analysis. We believed that our core–shell structure design and fast on-tip enrichment concept would gain more attention in PTM proteomes analysis in the near future.

## Methods

**Materials and reagents.** Hexadecyltrimethylammonium chloride (≥98%, CTAC), tetraethyl orthosilicate (98%, TEOS), 1,6-diisocyanatohexane (≥99.0%), tridecane (≥99%), ammonium hydroxide solution (28 wt%, NH$_3$·H$_2$O), ammonium fluoride (≥98%, NH$_4$F), 3- aminopropyltriethoxysilane (99%, APS), formaldehyde (37 wt% in water), sodium cyanoborohydride (NaBH$_3$CN), sodium cyanoborodeuteride (NaBD$_3$CN), trypsin (from bovine pancreas), protease inhibitor cocktail, formic acid (FA), urea, trifluoroacetic acid (TFA), dithiothreitol (DTT), iodoacetamide (IAA), and ammonium acetate were purchased from Sigma-Aldrich (St. Louis, MO, USA). Phosphatase inhibitor cocktail was ordered from Thermo Fisher Scientific (Rockford, USA). $^{13}$C-D$_2$-formaldehyde was purchased from Cambridge Isotope Laboratories, Inc. (Andover, USA). Dry pyridine was provided by Aladdin (99.8%, Shanghai, China). Toluene and chloroform (≥99.0%) were bought from Kaixin (Tianjin, China), and toluene was purified by refluxing (110 °C) over sodium and distilling (136 °C). Acetone (≥99.5%) and ethanol (≥99.8%) were obtained from Kermel (Tianjin, China). Acetonitrile (≥99.9%, ACN) was purchased from Merck (Darmstadt, Germany). Deionized water was used throughout the experiments (Millipore, Milford, MA, USA). Other reagents were all analytical grade. All the standard non-phosphorylated peptides were purchased from ChinaPeptides (Shanghai, China).

**Characterizations.** Hydrogen NMR ($^1$H-NMR) was measured by a bruker AVANCE III HD 400 MHz spectrometer at room temperature (Daltonios, Germany). Chemical shifts were recorded in parts per million (p.p.m.) using residual solvent peaks as internal references [CDCl$_3$ $\delta$: 7.26 (1H)]. Molecular mass was detected by orbitrap LTQ (Thermo, USA). TEM images were collected on a transmission electron microscope operated at 120 kV (JEM-2000EX, JEOL, Tokyo, Japan). SEM images were collected on a Zeiss Merlin FEG-SEM instrument. TGA was performed under an air atmosphere with a heating rate of 10 °C/min using a Netzsch STA449F3 thermogravimetric analyzer (Netzsch, Bavaria, Germany). Zeta potential was measured using the Nano-ZS90 zetasizer (Malvern, UK). X-ray photoelectron spectroscopy (XPS) characterization was carried out on an Thermo ESCALAB250Xi spectrometer with Al Kα radiation as the X-ray source (Thermo, Waltham, USA). The nitrogen adsorption and desorption isotherms were measured by QuadrasorbSI (Quadraorb, Wisconsin, USA). The Brunauer–Emmett–Teller (BET) method was used to calculate the specific surface areas with adsorption data in a relative pressure range from 0.049 to 0.252. Contact angle was measured by DSA100 (Krüss, Hamburg, Germany). MALDI-TOF spectra were recorded on Bruker Ultraflex III MALDI-TOF/TOF MS (Daltonios, Germany) and MALDI-TOF/TOF 5800 System (AB SCIEX, Foster City, CA) under positive reflection mode. ITC experiments were carried out on a MicroCal iTC200 instrument (Malvern,Sweden).

**Synthesis of Dpa (1)**. According to the literature[55], 2,2′-dipicolylamine (3.37 g, 16.9 mmol) and paraformaldehyde (0.813 g, 27.1 mmol) solution (water/i-PrOH = 5:3 (v/v), 48 mL) was adjusted to pH 8.0 by 1 N HCl aq. After stirring at 80 °C for 30 min, Boc-L-tyrosine-OMe (2.0 g, 6.77 mmol) was added, and the mixture was refluxed at 110 °C for 13 h. Then, the mixture was cooled to room temperature, and i-PrOH was removed by evaporation. After cooling on an ice-bath, the solution was removed by decantation, and the precipitated viscous oil was dissolved in 50 mL of AcOEt. The solution was washed with saturated NaHCO$_3$ and brine followed by drying over Na$_2$SO$_4$. After removal of the solvent in vacuo, the residue was purified by column chromatography (SiO$_2$, CH$_2$Cl$_2$/MeOH/ TEAB aq = 30/1/0.1 (v/v/v)) to give 1 (3.30 g, 68%) as a yellow oil. $^1$H-NMR (400 MHz, CDCl$_3$): δ 8.53 (4H, d, $J$ = 6 Hz), 7.62 (4H, td, $J$ = 7.5 Hz), 7.47 (4H, d, $J$ = 9 Hz), 7.13 (4H, td, $J$ = 6 Hz), 6.99 (2H, s), 5.23 (1H, d, $J$ = 6 Hz), 4.49–4.51 (1H, m), 3.85 (8H, s), 3.76 (4H, s), 3.59–3.61(3H, s), 2.99 (2H, s), 1.35 (9H, s). ESI-MS $m/z$ for C$_{41}$H$_{47}$N$_7$O$_5$ [M+H]$^+$ 718.35 and [M+Na]$^+$ 740.20 (Supplementary Figs. 15 and 16).

**Synthesis of amino terminal Dpa (2)**. In a typical reaction, TFA (20 mL) was added dropwise to a stirring solution of 1 (3.30 g, 4.60 mmol) in anhydrous CH$_2$Cl$_2$ (20 mL) on ice-bath, and the solution was stirred at room temperature for 2 h. After concentration in vacuo, the residue was dissolved in water and alkalized with ammonium hydroxide solution on ice-bath. The resulting mixture was extracted with CH$_2$Cl$_2$. The combined organic layers were washed with brine and dried over Na$_2$SO$_4$. After removal of the solvent in vacuo, 2 (2.86 g, 97%) was obtained as a pale viscous oil. $^1$H-NMR (400 MHz, CDCl$_3$): δ 8.53 (4H, d, $J$ = 6 Hz), 7.61 (4H, td, $J$ = 6 Hz), 7.46 (4H, d, $J$ = 9 Hz), 7.13 (4H, td, $J$ = 6 Hz), 7.03 (2H, s), 3.89 (8H, s), 3.81 (4H, s), 3.75–3.70 (4H, m), 3.68 (3H, S), 3.01–2.96 (1H, m), 2.80–2.75 (1H, m). ESI-MS $m/z$ for C$_{41}$H$_{45}$N$_7$O$_6$ [M+H]$^+$ 618.48, [M+Na]$^+$ 640.34 (Supplementary Figs. 15 and 17).

**Synthesis of carboxyl terminal Dpa (3)**. In a typtical reaction, glutaric anhydride (0.635 g, 5.57 mmol) was added to a solution of 2 (2.86 g, 4.64 mmol) in anhydrous CH$_2$Cl$_2$ (110 mL). The mixture was stirred and refluxed at 50 °C overnight. After removing the solvent by evaporation, 3 was obtained as a light yellow viscous oil. $^1$H-NMR (400 MHz, CDCl$_3$): δ 8.53 (4H, d, $J$ = 6 Hz), 7.61 (4H, td, $J$ = 7.5 Hz), 7.46 (4H, d, $J$ = 9 Hz), 7.13 (4H, td, $J$ = 6 Hz), 7.03 (2H, s), 3.86 (8H, s), 3.79 (4H, s), 3.70–3.65 (4H, m), 3.01–2.96 (1H, m), 2.80–2.75 (1H, m). ESI-MS $m/z$ for C$_{41}$H$_{45}$N$_7$O$_6$ [M+H]$^+$ 732.53, [M+Na]$^+$ 754.41 (Supplementary Figs. 15 and 18).

**Preparation of non-porous silica**. Non-porous microspheres were prepared by a seed-growth approach[56]. It contains three steps. Firstly, hydrolysis solution, containing 6.7 mL of NH$_3$·H$_2$O, 5.1 mL of H$_2$O, and 70 mL of CH$_3$CH$_2$OH, was placed in water bath at 22 °C, and 4.0 mL of TEOS was added. The mixtures were reacted for 40 min under constant mechanical stirring. After the temperature was increased to 55 °C, 0.64 mL of H$_2$O and 4.0 mL of pre-heated TEOS were added to the mixtures and reacted for 40 min (denoted as growth process), and the growth process was repeated another three times. The obtained suspension was denoted as seeds and divided into four pieces. Secondly, we used 1 piece seed replaced equal volume hydrolysis solution, and hydrolysis solution containing seed was heated to 55 °C, and the growth process was repeated three times. Afterward, the hydrolysis solution was added to the mixtures, and growth process was repeated four times. The obtained particles were centrifuged at 4000 × $g$ for 5 min, with 95% (v/v%) CH$_3$CH$_2$OH and H$_2$O. The obtained particles were divided to four pieces which were denoted as large particles. Finally, one piece of large particle was re-dispersed in the hydrolysis solution at 55 °C, and the growth process was repeated three times. Non-porous silica was obtained by centrifugation at 3300 × $g$ for 3 min using 95% (v/v%) CH$_3$CH$_2$OH and H$_2$O.

**Preparation of sub-2-μm core–shell silica**. Firstly, 1.0 g of above non-porous silica with were added to the growth system which included 100 mL of H$_2$O, 5.8 mL of tridecane, and 1.0 g of CTAC. Then, 26 mg of NH$_4$F and 6.0 mL of NH$_3$·H$_2$O were added and the mixtures were carried out for 24 h at 90 °C under stirring. The product was obtained by centrifugation at 3300 × $g$ for 3 min. Finally, the produce was dried at 65 °C for 2 h and calcined with a Ceramic Fibre Muffle Furnace (Michem, Beijing, China) at 550 °C for 6 h in air. The heating procedure was from 25 to 550 °C at a ramp rate of 1 °C/min. The pore size was enlarged by etching in 5 M HCl at 120 °C for 12 h.

**Amino-functionalized sub-2-μm superficially porous silica**. In a typical reaction, 0.6 g of sub-2-μm superficially porous silica particles and 2.21 mL of APTES were added to 30 mL of anhydrous toluene, and the mixtures were stirred and refluxed at 110 °C for 24 h. The obtained particles were centrifuged at 4000 × $g$ for 5 min and sequentially washed with toluene, CH$_3$OH, and CH$_3$COCH$_3$.

**Preparation of SiO$_2$@DpaZn**. In a typical reaction, a solution of carboxyl terminal Dpa 3 (750 mg, 1.1 mmol), amino-functionalized sub-2-μm superficially porous silica particles (600 mg), EDC (251 mg, 1.3 mmol), HOBt·H$_2$O (198 mg, 1.3 mmol), and DIEA (464 mg, 3.57 mmol) in dry DMF (21 mL) was stirred at room

temperature for 3 days. After filtration, the particles were subjected to sequential washing with toluene, methanol, and acetone. Then, bis-Zn (GA-DpaTyr-OMe)-sub-2-μm superficially porous silica particles (570 mg) were dissolved in water/ MeOH (1:1, v/v, 10 mL), and an water/MeOH solution (1:1, v/v, 20 mL) of Zn (NO$_3$)$_2$·6H$_2$O (2.97 g, 10 mmol) was added drop-wise to the silica particles suspension. After stirring at room temperature for 72 h, the obtained particles were centrifuged at 4000 × $g$ for 5 min, and sequentially washed with H$_2$O, CH$_3$CH$_2$OH, and acetone.

**NMR study**. $^1$H-NMR and $^{31}$P-NMR studies of DpaZn with pho-His were carried out on a Brucker ADVANCE III (500 MHz) at 25 °C. Samples were prepared using pho-His in D$_2$O in the absence or presence of DpaZn. TSP was used as an internal standard, and the observed peaks were individually assigned by the COSY experiments. In $^{31}$P-NMR experiment, 85% H$_3$PO$_4$ in H$_2$O was used as an external standard.

**Enrichment of N-phosphopeptide from BSA digests**. Five micrograms of standard N-phosphopeptide (TS$_p$HYSIMAR) and 500 μg of BSA digests were mixed and dissolved in 200 μL of 80% ACN (20 mM HEPES, pH 7.7). After incubation with 200 μg of SiO$_2$@DpaZn at room temperature for 20 min, total volume of 400 μL 80% ACN, 400 μL 65% ACN, and 400 μL 0.001% NH$_3$·H$_2$O comprised washing steps, respectively. Phosphopeptides were eluted with 100 μL of 0.1% NH$_3$·H$_2$O and analyzed by MALDI-TOF MS.

**Adsorption kinetics of SiO$_2$@DpaZn**. A total of 0.4 mg of SiO$_2$@DpaZn was incubated with 1.6 mL of 80% ACN (20 mM HEPES, pH 7.7) containing 10 μg of O-phosphopeptide GK8 at room temperature. After incubation for 2, 5, 10, 15, 20, and 30 min, respectively, 40 μL of the mixture was centrifuged, and the concentration of GK8 was measured by UV analysis. The adsorption capacity ($Q$) was calculated as follows: $Q = (C_0 - C)\,V/m$, where $m$ is the mass (mg) of SiO$_2$@DpaZn, $V$ (mL) the volume of incubation, $C_0$ (mg/mL) and $C$ (mg/mL) the concentrations of GK8 in the initial solution and the supernatant after the adsorption, respectively. Three sets of parallel experiments were conducted simultaneously.

**On-tip enrichment of phosphopeptides by SiO$_2$@DpaZn tips**. A total of 1 mg of SiO$_2$@DpaZn in 80% ACN (20 mM HEPES, pH 7.7) was packed into a 20-μL micropipette tip by centrifugation at 5400 × $g$ for 10 min, and then equilibrated with 10 μL of 80% ACN (20 mM HEPES, pH 7.7). Sample loading, washing, and elution were realized by centrifugation at 5400 × $g$ for 5, 12, and 5 min, respectively. Dissolved in 20 μL of 80% ACN, peptides were load onto SiO$_2$@DpaZn tips. After washing three times with 80 μL of 50% ACN (20 mM HEPES, pH 7.7), 80 μL of 0.001% NH$_3$·H$_2$O, and 80 μL of 0.005% NH$_3$·H$_2$O orderly, samples were eluted with 10 μL of 1% NH$_3$·H$_2$O. Eluates were analyzed by MALDI-TOF MS.

**Preparation of protein extracts from E. coli**. E. coli (strain K12) was cultured in Luria–Bertani medium (LB, 5 g/L yeast extracts, 10 g/L NaCl, and 10 g/L tryptone) at 37 °C overnight or in M9 minimal medium (consisting of 6 g/L Na$_2$HPO$_4$, 3 g/L KH$_2$PO$_4$, 0.5 g/L NaCl, 1 g/L NH$_4$Cl, supplemented with either additional 0.5% (w/ v) glucose or 0.5% glycerol) at 37 °C to exponential phase and stationary phase, and were harvested by centrifugation at 4000 × $g$ for 2 min at 4 °C. Cells were washed three times with cold PBS and resuspended in lysis buffer (8 M urea, 10 mM PBS, 1% (v/v) protease inhibitor cocktail, 10 mM EDTA, and 1% (v/v) phosphatase inhibitor cocktail). After ultrasonication for 120 s (20 s interval every 10 s) in an ice bath, insoluble portions were separated from the soluble ones by centrifugation at 16,000 × $g$ for 30 min at 4 °C, and protein concentrations were determined using BCA assay.

**Tryptic digestion of E. coli**. E. coli protein extracts were denatured at 95 °C for 1 min. Then, the disulfide bond was reduced with 25 mM DTT at 37 °C for 75 min and the resulting cysteine residues were alkylated with 50 mM IAA at room temperature in the dark for 30 min. The excess IAA was quenched with 50 mM DTT. The resulting mixture was diluted with 20 mM ammonium bicarbonate to the final concentration of urea <1 M. Afterward, trypsin (1:3) was added and incubated at 37 °C for 1.5 h. The resultant peptides were desalted by RP-LC.

**Preparation and digestion of protein extracts from mammalian cells**. The HeLa and HEPG2 cells were cultured in DMEM medium containing 10% FBS and 1% PBS in a humidified incubator of 5% CO$_2$ and 95% air at 37 °C. All cells reaching 80% confluence were detached with 0.25% trypsin/EDTA and centrifugated (540 × $g$, 5 min) to collect the cells. Cells were washed three times with cold PBS and resuspended in lysis buffer (8 M urea, 10 mM PBS, 1% (v/v) protease inhibitor cocktail, 10 mM EDTA, and 1% (v/v) phosphatase inhibitor cocktail). After ultrasonication for 120 s (20 s interval every 10 s) in an ice bath, insoluble portions were separated from the soluble ones by centrifugation at 16,000 × $g$ for 30 min at 4 °C, and protein concentrations were determined using BCA assay. After treating with DTT and IAA, the proteins were precipitated

with CHCl₃/CH₃OH. Afterward, trypsin (1:3) was added and incubated at 37 °C for 1.5 h.

**Enrichment of *E. coli* and HeLa cells phosphopeptides by SiO₂@DpaZn tips**. A total of 36 mg of SiO₂@DpaZn in 80% ACN was packed into 200-μL micropipette tips equally by centrifugation at 5400 × g for 10 min. Disolved in 200 μL of 80% ACN, 12 mg of *E. coli* digests were loaded onto SiO₂@DpaZn tips by centrifugation at 5400 × g for 10 min. Washing steps were carried out with 100 μL of 50% ACN, 100 μL of 30% ACN, 100 μL of 0.001% NH₃·H₂O, and 80 μL of 0.005% NH₃·H₂O, respectively, and phosphopeptides were eluted with 20 μL of 1% NH₃·H₂O for two times. Elutes was mixed up for the following high pH-RP fractionation. For mammalian cell phosphopeptides, 30 mg of SiO₂@DpaZn was used for 6 mg of cell lysate digests enrichment.

**MALDI-TOF-MS**. One microliter of the sample was deposited on the sample plate and dried, then 1 μL of 20 mg/mL DHB substrate (in 60% (v/v) acetonitrile and 0.1% (v/v) TFA aqueous solution) was introduced. MS spectra were acquired on an Ultraflex III TOF/TOF mass spectrometer (Bruker Daltonik GmbH, Germany) equipped with a 200-Hz smart-beam 1 laser at 355 nm or a MALDI-TOF/TOF 5800 System (AB SCIEX, Foster City, CA) quipped with a 1-kHz optibeam™ on-axis laser in the positive reflection mode, and controlled by the Flex Control 2.4 software and Data Explorer software 4.11. Flex Analysis software 3.3 and TOF/TOF Series Explorer software 4.1 were used to analyze MALDI-TOF MS data. The threshold for peak acceptance was a signal-to-noise ratio of 6.

**NanoRPLC-ESI-MS/MS methods**. For *E. coli* lysate digests analysis, each fraction of the enriched phosphopeptides was automatically loaded onto a RP trap column (150 μm i.d. × 5 cm) and separated by a C18 capillary column (150 μm i. d. × 15 cm). The trap column and the analytical column were both packed in-house with 5 μm, 100 Å Venusil XBP C18 silica particles. In order to study the phosphorylated peptides in *E. coli* cultured in LB medium, two mobile phases (A: 2% (v/v) ACN with 0.1% (v/v) FA and B: 98% (v/v) ACN with 0.1% (v/v) FA) were used to generate a 75-min gradient with the flow rate of 500 nL/min (45 min from 7 to 25% B, 20 min from 25 to 40% B, 5 min from 40 to 80% B, and 5 min to kept at 80% B). A Q-Exactive mass spectrometer (Thermo-Fisher, San Jose, CA, USA) was operated in full scan (70,000 FWHM, 350–1800 m/z) and product scan (17,500 FWHM, 100–1000 m/z) modes at positive ion mode. The electro-spray voltage was 2.3 kV, and the heated capillary temperature was 270 °C. All the mass spectra were recorded with Xcalibur software (version 3.1, Thermo Fisher Scientific, USA). MS/MS spectra were acquired by data-dependent acquisition mode, and the 20 most intense peaks with charge state ≥2 were selected for sequencing in the HCD collision cell with stepped collision energy of 28%, 30%, and 32%. Each fraction was analyzed in duplicate. Furthermore, in order to study the phosphorylated peptides in *E. coli* cultured in M9 medium, two mobile phases (A: 0.1% (v/v) FA in water and B: 98% (v/v) ACN with 0.1% (v/v) FA) were used to generate a 121-min gradient with the flow rate of 600 nL/min (90 min from 7 to 25% B, 20 min from 25 to 40% B, 1 min from 40 to 80% B, and 10 min to kept at 80% B). A Orbitrap Fusion Lumos mass spectrometer (Thermo-Fisher, San Jose, CA, USA) was operated in full scan (60,000 FWHM, 350–1500 m/z) and product scan (15,000 FWHM, 100–1000 m/z) modes at positive ion mode with orbitrap detector. The electro-spray voltage was 2.3 kV, and the heated capillary temperature was 320 °C. All the mass spectra were recorded with Xcalibur software (version 3.1, Thermo Fisher Scientific, USA). MS/MS spectra were acquired by data-dependent acquisition mode, and total cycle time was set to 3 s. Peptides with charge state ≥2 were selected for sequencing in the HCD collision cell with collision energy of 30%. HPLC separation for Hela lysate is as same as that for *E. coli* cultured in M9 medium, and an Orbitrap Fusion Lumos mass spectrometer (Thermo Fisher, San Jose, CA, USA) was operated in full scan (120,000 FWHM, 350–1500 m/z) mode at positive ion mode with orbitrap detector. The electro-spray voltage was 2.5 kV, and the heated capillary temperature was 320 °C. All the mass spectra were recorded with Xcalibur software (version 3.1, Thermo Fisher Scientific, USA). MS/MS spectra were acquired by data-dependent acquisition mode, and total cycle time was set to 3 s. Peptides with charge state ≥2 were selected for sequencing with HCD (collision energy 32%, max injection time 35 ms) and neutral-loss-triggered (Δ97.98) EThcD (ETD reaction time 50 ms, max ETD reagent injection time 200 ms, supplemental activation energy 25%, max injection time 35 ms) for fragmentation. All product ions were detected in the ion trap (rapid mode).

**Database search**. Proteome Discoverer 2.1 (for analysis of the phosphorylated peptides in *E. coli* cultured in LB medium) with MASCOT 2.4 and Maxquant 1.6.0 (for the phosphorylated peptides in *E. coli* cultured in M9 medium and HeLa) were applied for database searching against the NCBI_E.coli_k12 database (updated on 01/18/2016, 4127 proteins) or Uniprot_Homo Sapiens database (updated on 04/04/2019, 42432 proteins). The corresponding reversed database was also performed to evaluate the false discovery rate (FDR) of peptide identification in the database searching process. The parameters of database

searching included: up to three missed cleavages allowed for full tryptic digestion, precursor ion mass tolerance 10 p.p.m., product ion mass tolerance 20 m.m. u., carbamidomethylation (C) as a fixed modification, and oxidation (M), phosphorylation (STYHRK), acetyl (protein N-term), deamidated (NQ) as variable modifications. PSMs were validated using perculator based on q-value at a 1% FDR. GO analysis was carried out on http://pantherdb.org/ (Panther 15.0) and http://revigo.irb.hr/. Sequence motif analysis was carried out on Weblogo 2.8.2.

**ITC experiment**. ITC titration was performed on an Isothermal Titration Calorimeter from MicroCal Inc. All measurements were conducted at 298 K. In general, a solution of the peptide (1–2 mM) in 50 mM HEPES buffer (pH 7.2) was injected stepwise (10 μL × 24 times) into a solution of DpaZn (25–100 μM) dissolved in the same solvent system. The measured heat flow was recorded as function of time and converted into enthalpies (ΔH) by integration of the appropriate reaction peaks. The binding parameters (K_D, ΔH, ΔS, n) were evaluated by applying one site model using the software Origin (MicroCal Inc.).

**Reporting summary**. Further information on research design is available in the Nature Research Reporting Summary linked to this article.

## Data availability

The authors declare that all data supplementary to the findings of this study are available within the paper and its supplementary information or from the corresponding author upon reasonable request. Proteomics data have been deposited to the ProteomeXchange Consortium via the iProX partner repository with the dataset identifiers PXD017423 (for LB and different carbon source cultured *E. coli*) and PXD021067 (for HeLa and HEPG2).

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

## Acknowledgements

The authors are grateful for the finanical support from National Key R&D Program of China (2017YFA0505003 and 2016YFA0501401), National Natural Science Foundation (21505133, 91543201, 91753110, and 21725506), and CAS Key Project in Frontier Science (QYZDY-SSW-SLH017).

## Author contributions

Y.Z., L.Z., and B.J. conceived and designed the project. Y.H. and B.J. synthesized the materials and carried out the characterization. Y.W. and B.Z. performed the standard pLys peptide enrichment. Y.C. performed the NMR experiments and L.L. collected ITC data. Q.W. and Z.L. extracted proteins from *E. coli*. B.J. and Z.S. extracted proteins from mammalian cells. All authors participated in the data analysis, discussed the results, and took part in producing the manuscript.

## Competing interests

The authors declare no competing interests.
