## [Peer Review File · Nature Communications]

Reviewers' comments:

Reviewer #1 (Remarks to the Author):

The manuscript described enrichment method of N-phosphorylated peptide from biological sample by using Zn-complex modified silica particle. The method seems to enrich N-phosphorylated peptide from proteolytic sample fairly well and actually they found N-phosphorylated site in some proteins with this method. Therefore, the method is curious to determine the role of N-Phosphorylation in biological systems. Thus, the manuscript will meet the requirement of publication in the journal. However, I think following points should be reconsidered and revised before publication.

- 1) The zinc-complex also traps O-phosphorylated peptides which should exist larger amounts and more stable. However, number of enriched O-phosphorylated peptides in data S1 is much smaller than that of N-phosphorylated peptides. Why the particle binds N-phosphorylated peptides preferentially against O-phosphorylated peptides?
- 2) Kd values of the particle for N-phosphorylated peptides ranged from 10-100 μ M. This binding ability seems to be too small to enrich the small amount of phosphorylated peptide in proteome mixture that should contain various amounts of phosphorylated peptides, because common cell lysate can't contain such high concentration of every target peptide.
- 3) There have been reported so many systems to enrich O-phosphorylated peptide such as Ga-complexed particle, Phos-tag related complex modified particle and so on. They should compare the performance of this system with other systems for O-phosphorylation enrichment. Otherwise, they should clarify why the system prefers N-phosphorylated peptide compared with other systems.
- 4) In line 88, Fig. S3 doesn't indicate the sentence explanation.

Reviewer #2 (Remarks to the Author):

The manuscript by Yechen Hu and coworkers describes a new Bis(zinc(II)-dipicolylamine) functionalized micro-beads for specific enrichment of N-phosphorylated peptides enabling large-scale N-phosphoproteomics.

Large-scale phosphoproteomics has so far mainly focused on analyzing the more abundant serine, threonine and tyrosine O-phosphorylation sites, but N-phosphorylation on arginine, histidine and lysine residues is now emerging as a widespread modification in both bacteria and eukaryotes. However, large-scale N-phosphoproteome analysis is challenging due to technical issues, i.e. N-phosphosites are labile at low pH and therefore lost during standard phosphoproteomics sample preparation procedures, which makes use of enrichment under acidic conditions.

Recently few labs - Heck, Becker and Mechtler labs, respectively - have demonstrated that optimizing enrichment procedures at higher pH allow them to preserve N-phosphorylation sites. However, no specific enrichment of N-phosphopeptides over O-phosphopeptides are currently available hampering large-scale analysis. The introduction of Bis(zinc(II)-dipicolylamine) functionalized micro-beads could solve this problem.

Using the new micro-beads the authors demonstrate the N-phosphopeptide enrichment efficiency and specificity using synthetic N-phosphopeptides and continue to validate their method analyzing N-phosphorylated peptides from E. coli lysates. However, this important part of the manuscript is not convincing to me. The authors need to do a better job in describing and presenting the large-scale E. coli N-phosphoproteomics screen, which is the major selling point of the manuscript and benchmark their method against current state-of-the-art. Moreover, since the raw MS-data is not made available it is impossible for me (and the community in the future) to assess the quality of the dataset and the claims made. Consequently, I cannot recommend publication of the manuscript in its current form as the phosphoproteomics dataset simply does not keep up to the

community standards.

MAJOR POINTS:

Firstly, lack of availability of raw MS data. No raw MS data/output files are provided for download for the reviewers, which makes it impossible for other researchers to evaluate and scrutinize the quality of the claimed results. The raw LC-MS/MS data files and the associated search files and output tables should be made available to the proteomics community, for example, through the MASSIVE or ProteomeXchange repositories. Simply stating that all relevant data supporting the findings of this study are available from the corresponding authors upon request is not acceptable.

Secondly, in addition to making the raw MS data and results available, the material and methods around parameters of MS analysis is also incomplete as it is completely unclear how high pH fractionation was employed, how many fractions were analyzed and which MS settings employed. This needs to be addressed.

Thirdly, the presented study is not the first one to investigate the E.coli N-phosphoproteome. To demonstrate the claimed superiority of the Bis(zinc(II)-dipicolylamine) functionalized micro-beads for specific enrichment of N-phosphorylated peptides the authors should benchmark their methods and overlap the phosphopeptides identifications against the E.coli His-phosphoproteome dataset from the Heck lab (Potel et al, Nature Methods 2018).

Fourthly, for the method to be useful for studying N-phospho-signaling dynamics the authors need to demonstrate the reproducibility of the enrichment procedure. How well does the MS signal intensities of all identified peptides correlate between replica?

Finally, complete supplementary tables with all identified peptides (both phosphor- and non-phospho) with all associated quality metrics and MS signal intensities should be provided for all the different E.coli high pH fractions analyzed.

Reviewer #3 (Remarks to the Author):

Here the authors set out to develop an improved method for enriching phosphopeptides containing an N-linked phosphate, such as pHis, pLys or pArg, from a tryptic digest, which is currently difficult to do since phosphoramidate linkages are unstable under conventional acidic phosphopeptide enrichment conditions. For this purpose, they covalently attached bis(zinc(II)-dipicolylamine) groups, which chelate protein/peptide linked phosphate residues, to sub-2 μm core-shell silica microspheres, to generate a SiO₂@DpaZn matrix. Using a synthetic TSpHYSIMAR peptide, originating from BSA, they showed that SiO₂@DpaZn beads were able to efficiently capture synthetically phosphorylated pHis-containing peptides under neutral pH 7.7 conditions within 30 min, and that the pHis phosphopeptides could be eluted intact with 1% NH₃, for subsequent MS analysis. They went on to demonstrate that SiO₂@DpaZn bead capture of pHis peptides was more efficient than solution capture. Using their new method, the authors were able to identify 40 pHis sites in 32 proteins in digests of proteins in an E. coli lysate.

As demonstrated here, DpaZn groups linked to SiO₂ beads can enrich pHis peptides, but there is no evidence that they selectively enrich peptides with P-N linkages. Surprisingly, the authors did not determine whether SiO₂@DpaZn beads also enrich pSer, pThr or pTyr peptides, and parallel analysis of synthetic pSer, pThr and pTyr peptides needs to be carried out (for this purpose, the same BSA peptide, which contains all three hydroxyamino acids, could be used for generation of pSer/pThr/pTyr phosphopeptides by standard chemical synthesis). Information as to whether the beads also enrich pSer/pThr/pTyr peptides might be evident from their MS data of phosphopeptides enriched on SiO₂@DpaZn beads from a tryptic digest of E. coli cells (although these cells have only low levels of pSer, pThr and pTyr proteins). In fact, the list of identified

phosphopeptides in Data S1 show clearly that pSer, pThr, and pTyr peptides were enriched, but there is no mention of this in the paper! In general, the authors need to do a better job of discussing what published technologies are already used to enrich pHis and other P-N peptides and carry out MS analysis, and ideally compare their method with such published methods (in this regard the authors might want to look at a recent BioRxiv paper from Claire Evers' group (<https://doi.org/10.1101/202820>), which reports another neutral pH method for enrichment of peptides containing non-canonical (and canonical) phosphoresidues).

In summary, this new method for enriching tryptic peptides containing pHis, pLys and pArg has potential, particularly because it uses neutral/alkaline pH and the pHis peptides are bound to the SiO₂@DpaZn beads very rapidly, but these studies suffer from a lack of key controls, deficiencies in the MS analysis, and in the end it is not clear whether the enrichment is selective for peptides containing a phosphoramidate linkage as the authors claim, or whether it enriches all phosphopeptides as one might expect. More work needs to be done to validate this method, show that it is superior to other methods by direct comparison, and determine how selective it is for the N-phosphoproteome.

Points: 1. Abstract/Page 2, line 9: As written, it is not clear what the N-phosphopeptides were being selected from nor what the 44.1% value represents. Was this enrichment from total E. coli lysate peptides considering total phospho-STYHKR peptides, or based only on recovery of BSA (TSHpYSIMAR) as shown in Figure 2d?

2. Page 3: It is unclear whether SiO₂@DpaZn beads enrich all types of phosphopeptide or whether they are specific for phosphoramidate bond containing peptides as the authors imply. Apparently, the main reason why the SiO₂@DpaZn bead method is better for N-phosphopeptides is because of the compatibility with alkaline/neutral pH. But in that case, it would not eliminate SiO₂@DpaZn bead binding to other types of phosphopeptides, and therefore the authors need to consider more than N-phosphates.

3. Page 4, bottom: The convention in writing out peptides containing a phosphoresidue is to use pS, pT, pY, pH, etc. As it stands, by using TSHpYSIMAR, it is unclear whether it is the Tyr or the His residue in the TSHHYSIMAR peptide that is phosphorylated! This needs to be changed throughout the paper.

4. Page 4: How did the authors rule out the possibility that the Arg in this peptide was also phosphorylated during the chemical phosphorylation reaction with potassium phosphoramidate - MALDI-TOF MS analysis alone would not discriminate between the pHis and pArg forms. In addition, although perhaps less likely, how did they rule out the presence of phosphate on the Ser, Thr and Tyr residues in the chemically phosphorylated peptide? This would require MS/MS analysis to define the b/y ions.

5. Page 4: Were the authors able to measure the relative levels of the phosphopeptide versus the unphosphorylated form of the peptide before enrichment?

6. Figure 2: Since there was no peptide fragmentation in this analysis, the authors should state that they obtained a phosphopeptide with a single phosphate (+80 Da) and made the assumption that it is an N-phosphate solely based on the enrichment conditions, N.B. the neutral/alkaline conditions used here do not cause degradation of pS/pT/pY and so there is no reason to think that N-phosphate peptides were selectively retained. Oslund et al. (ref 35) and Potel et al. (ref 39) used the same peptide sequence as a control, but showed that there was also a pTyr version (debate about whether the "triplet" neutral loss fingerprint could differentiate the pHis from the pTyr).

7. Page 6/Figure 2: The speed of enrichment is definitely a major aspect/advantage of the

SiO₂@DpaZn bead method, but the justification for this experiment is not obvious. It is not standard to vortex or sonicate cell extracts, even for recovery of pS/pT/pY proteins. Presumably this is intended to solubilize proteins that would otherwise pellet even though extraction was done in 8 M urea. But if the authors really want to compare the impact of vortex and sonication, they should indicate the speed of vortex and the power of sonication (watts/joules).

8. Page 6: For a standard analysis with MaxQuant, the cut-off is usually set at 40. So the level of 20 used here seems very low, knowing that they are generally recalibrated below 70, and 20 seems somewhat arbitrary.

9. Page 6/Data S1: The authors need to discuss the fact that they also found pSer/pThr/pTyr-containing peptides among the tryptic peptides enriched from the digest of the *E. coli* lysate, and indicate that this means that the SiO₂@DpaZn beads do not only enrich for peptides with N-phosphates. With regard to the identified pArg and pLys peptides, many of them have a C-terminal pLys or pArg, but since trypsin should not cleave at pArg or pLys it is unclear how these peptides could have been generated or how these assignments could have been made. In terms of the identities of the pHis sites, several appear to be active site pHis residues used for phosphate transfer (e.g. phosphoglucosyltransferase), but did any of them correspond to the active site pHis in two-component His kinases?

10. The authors could do a better job of comparing the pHis, pLys and pArg-containing *E. coli* protein-derived peptides they identified with those reported in the Oslund et al. and Potel et al. papers.

11. Page 7: It is not clear whether the BSA peptide with a single phosphate on Ser2 would in fact exhibit a major difference in retention time versus a mono-pHis form of the peptide. They could compare the chemically phosphorylated pHis peptide with a synthetic pSer version of the peptide to establish this. Only a good coverage of the b/y ion series from fragmentation would be able to distinguish the two sites, since the mass and the charge are similar

12. Page 7, bottom: All three of these amino acids can be phosphorylated (two generate non-conventional phosphoamino acids that could potentially be enriched and detected under the neutral/alkaline conditions used here), but apparently the authors did not consider pCys and pAsp in their *in silico* interrogation.

13. Page 9: Once again, it is not clear whether SiO₂@DpaZn bead enrichment is specific for N-phosphate, or specific to peptide-linked phosphate in general, working on N-phosphate as well.

14. Page 13: Only MALDI-TOF MS was used for synthetic phosphopeptide characterization, and so authors cannot validate the position of the phosphate enriched in their peptides.

15. Page 14, line 7: This implies that only MALDI-TOF MS was used to characterize the eluted peptides, but at the bottom of the page it states that NanoRPLC-ESI-MS/MS on a Q-Exactive MS system was used. This needs to be clarified, and a description added for exactly how the existence and position of the phosphate in enriched peptides were established.

16. Page 14: The precise power (watts/joules) needs to be included.

17: Page 14: The *E. coli* protein extracts were denatured at 95°C for 1 min in 8 M urea and PBS (presumably around pH 7, and one would expect this heating to lead to significant loss of phospho-N bond phosphate, which is a concern,

18. Page 14: The use of trypsin at a 1:3 ratio with substrate protein is much higher than is typical (1:100), and runs the risk of generating non-tryptic cleavages..

19. Page 15, top: The authors need to indicate the pH of the buffers used for the C18 HPLC separation of peptides. Since the buffers contained formic acid they will presumably have been ~ pH 2, which will also lead to loss of phospho-N bond phosphate. In this regard, did the authors identify any nonphosphorylated peptides corresponding to known sites of His phosphorylation,, which might represent pHis peptides that were successfully enriched but then lost their phosphate during the HPLC fractionation step.

20. Page S12: Twenty is a low cutoff score - generally 40 is a standard; did the authors consider the probability of localization when the scores are different? If the scan number is different, how do they consider the fact that both sites can be phosphorylated on different related phosphopeptides with different scores?

Response to the comments of Reviewer 1

Question 1: The zinc-complex also traps O-phosphorylated peptides which should exist larger amounts and more stable. However, number of enriched O-phosphorylated peptides in data S1 is much smaller than that of N-pho peptides. Why the particle binds N-Pho preferentially against O-pho peptides?

Reply: We do appreciate your kindly comment. We also found the zinc-complex trap all type phosphorylated peptides including O-phosphorylated peptides. According to previous research (J. Am. Chem. Soc., 126, 2454-2463 (2004)), the DpaZn(II) coordination complex has a vacancy on each Zn^{2+} ions and the phosphate anion $[RO(N)PO_3^-]$ can access to form the complex $RO(N)PO_3$ -DpaZn(II) under neutral condition. As shown in reply Figure 1, electron donating ability of the N atom is stronger than that of O atom, which results in more stable structure of $RNPO_3$ -DpaZn(II). It may lead to the N-Pho preferentially against O-pho peptides under the certain amount of material. The description was added in the revised manuscript at discussion section. Furthermore, N-phosphorylations, especially pHis, play important roles in the he main regulatory and signaling systems in bacteria: two-component systems (TCS) and the phosphoenol pyruvate: carbohydrate phosphotransferase (PTS) systems (Arch. Microbiol. 181, 171–181(2004), Mol. Cell. Proteomics, 13, 537-550 (2014)). Compared to O-phosphorylation, there may be higher abundance N-phosphorylated proteins involve in the life of *E. coli*, allowing for perception and response to environmental changes. Therefore, number of enriched O-phosphorylated peptides in data S1 is smaller than that of N-pho peptides. Similar results were found in the different carbon resource *E. coli* (data S2).

Figure 1. Schematic representation of the interaction between DpaZn(II) and

phosphorylated peptide. Electron donating ability of the N atom in P-N is stronger than that of O atom in P-O, which results in more stable structure of RNPO₃-DpaZn(II).

Question 2: K_d values of the particle for N-pho peptides ranged from 10-100 uM. This binding ability seems to too small to enrich the small amount phosphorylated peptide in proteome mixture that should contains various amount of phosphorylated peptides, because common cell lysate can't contain such high concentration of every target peptides.

Reply: We are grateful for your comment. We must apologize for our previous unclear statement. In our manuscript, K_D value represented dissociation constant which was obtained by isothermal titration calorimetry (ITC) experiment. In our ITC experiments, N-phosphopeptides (for example, 30 μM Peptide-a) were dropped in a solution contain 40 μM DpaZn, in which DpaZn was uniformly distributed. However, just as the immobilized enzymatic reactor can condense the concentration of enzyme and enhance the digestion efficiency (Analytical Chemistry, 89, 6324-6329 (2017)), in real enrichment of phosphopeptides from *E.coli* lysates, 3 mg of our SiO₂@DpaZn(II) was tightly packed into a 200 μL micropipet tip to achieve a volume of just 3.8 μL. This will increase at least 100-fold for enrichment ability. It can recognize N-pho peptides as low as nmol or even lower. The copy number of most identified N-phosphoproteins is less than 1000 (Data S1). In conclusion, our on-tip enrichment method is more suitable for low-abundance target enrichment.

Question 3: There have been reported so many system to enrich O-phosphorylated peptide such as Ga-complexed particle, Phos-tag related complex modified particle and so on. They should compare the performance of this system with other systems for O-phos enrichment. Otherwise, they should clarify why the system prefers N-phos peptide comparing with other systems.

Reply: We do appreciate your kindly suggestion. Actually, O-phosphoproteome has achieved great breakthroughs with the development of novel affinity materials

(Mol. Cell. Proteomics, 7, 661-671 (2008); 1162-1165 (2016)). However, they are performed under strong acidic conditions which are not suitable for N-phosphoproteome analysis. In our work, we aimed to develop a method friendly to N-phosphorylated peptides under neutral condition (also friendly to O-phosphorylated peptides; but is not our main purpose). We explained the reason why the system prefers N-phos peptide comparing with other systems in question 1.

Question 4: In line 88, Fig. S3 doesn't indicate the sentence explanation.

Reply: We do appreciate your kindly reminder. We mislabeled the supporting figures with the explanation in manuscript. It should be Fig. S4, and the following numbers were also revised.

Response to the comments of Reviewer 2

Question 1: Firstly, lack of availability of raw MS data. No raw MS data/output files are provided for download for the reviewers, which makes it impossible for other researchers to evaluate and scrutinize the quality of the claimed results. The raw LC-MS/MS data files and the associated search files and output tables should be made available to the proteomics community, for example, through the MASSIVE or ProteomeXchange repositories. Simply stating that all relevant data supporting the findings of this study are available from the corresponding authors upon request is not acceptable.

Reply: Thank you for your kindly suggestion, the mass spectrometry proteomic data of *E.coli* (contains raw LC-MS/MS data files and the associated search files and output tables) were deposited to the ProteomeXchange Consortium via the PRIDE partner repository with the dataset identifier. The mass spectrometry proteomics data have been deposited to the ProteomeXchange Consortium (<http://proteomecentral.proteomexchange.org>) via the iProX partner repository with the dataset identifier PXD017423.

Question 2: in addition to making the raw MS data and results available, the material

and methods around parameters of MS analysis is also incomplete as it is completely unclear how high pH fractionation was employed, how many fractions were analyzed and which MS settings employed. This needs to be addressed.

Reply: We are grateful for your kindly reminding. The enrichment methods, MS parameters and high pH fractionation information were provided in the revised manuscript and revised supporting information, respectively.

Question 3: the presented study is not the first one to investigate the E.coli N-phosphoproteome. To demonstrate the claimed superiority of the Bis(zinc(II)-dipicolylamine) functionalized micro-beads for specific enrichment of N-phosphorylated peptides the authors should benchmark their methods and overlap the phosphopeptides identifications against the E.coli His-phosphoproteome dataset from the Heck lab (Potel et al, Nature Methods 2018).

Reply: We do appreciate your kindly suggestion. In Potel's work, there were 135 credible pHis sites identified from two special cultured *E.coli* cells (M9 minimal medium, consisting of M9 salts (6 g/L Na₂HPO₄, 3 g/L KH₂PO₄, 0.5 g/L NaCl, 1 g/L NH₄Cl) supplemented with either additional 0.5% (w/v) glucose or 0.5% glycerol). It should be noticed that phosphorylation sites, including N-phosphorylation sites, were highly accumulated in such mediums (J. Proteome Res., 12, 2611-2621(2013); Nat. Biotechnology, 34, 104-113(2016)). In our previous results, 11 credible pHis sites from 13 pHis peptides were identified from normal cultured *E.coli* cells (Luria-Bertani medium, consisting of 5 g/L yeast extracts, 10 g/L NaCl and 10 g/L tryptone). Actually, it is closer to the actual situation of N-phosphorylation in bacteria under such condition. We found 7 pHis sites from 9 pHis peptides were identified by two methods.

Furthermore, N-phosphopeptides from glucose and glycerol cultured *E.coli* (exponential phase and stationary phase) were used as samples for our method. After searching through Maxquant software (to compare with literature), setting the score over 40 and score difference over 5, 162 N-phosphopeptides (20 pHis, 67 pLys and 75 pArg) with localization probability over 0.75 were identified. Comparing the 135

credible pHis identified by Potel's, our method shows great complementarity in providing information of pLys and pArg peptides. There are two reasons to explain the different results between our and Potel's work. On the one hand, in addition to the standard sample pretreatment steps, protein precipitation and benzonase treatment were implied before digestion which significantly reduce the inference of pollutants. On the other hand, pH 2.3 was used in the enrichment processes which increase the affinity ability of Fe³⁺-IMAC. These two processes no doubt increase the enrichment ability of pHis. However, considering pLys and pArg peptides, especially pLys peptides (Anal. Bioanal. Chem., 411, 4159-4166 (2019); Sci. China Chem., 62, 708-712 (2019)) are suffered from severe hydrolysis under long operation and the acidic condition. Our method might be more suitable for pLys and pArg peptides enrichment. Therefore, according to the different experimental purposes, you need to choose the appropriate methods.

Question 4: for the method to be useful for studying N-phospho-signaling dynamics the authors need to demonstrate the reproducibility of the enrichment procedure. How well do the MS signal intensities of all identified peptides correlate between replica?

Reply: We do appreciate your kindly suggestion. After label-free quantification by the MaxQuant software, calculated Pearson correlation coefficients (FCC) for the quantified peptides in triplicate (each replicate corresponding to a different enrichment and different LC-MS/MS injection) were all around 0.9 (Figure 2). This shows that our enrichment method is robust and reproducible (Nat. Methods, 15, 187–190 (2018)). This result was also added to the SI of revised manuscript on revised supporting information Fig. S11.

Figure 2. Assessment of the reproducibility of enrichment procedures

Question 5: complete supplementary tables with all identified peptides (both phosphor- and non-phospho) with all associated quality metrics and MS signal intensities should be provided for all the different E.coli high pH fractions analyzed.

Reply: We do appreciate the reviewer's kindly suggestion. All identified phosphopeptides with all associated quality metrics and MS signal intensities have been provided in the Data S1 and S2. The selectivity of the material is not too high for complex biological samples due to neutral condition. So we also identified abundant non-phosphopeptides in our results, which was inevitable for complex biological sample. However, it made more sense to provide phosphorylated protein data.

Response to the comments of Reviewer 3

Question 1: Abstract/Page 2, line 9: As written, it is not clear what the N-phosphopeptides were being selected from nor what the 44.1% value represents. Was this enrichment from total E. coli lysate peptides considering total phospho-STYHKR peptides, or based only on recovery of BSA (TSHpYSIMAR) as

shown in Figure 2d?

Reply: Thank you for your kindly suggestion. We firstly apologized for our inaccurate expression. The N-phosphopeptide mentioned here is a standard pHis peptide (TSpHYSIMAR) by chemical synthesis. The value 44.1% represented the recovery of TSpHYSIMAR by on-tip enrichment method. It neither represented enrichment from total E. coli lysate peptides considering total phospho-STYHKR peptides nor base only on recovery of BSA. The detail measures were described in supporting information.

Question 2: Page 3: It is unclear whether SiO₂@DpaZn beads enrich all types of phosphopeptide or whether they are specific for phosphoramidate bond containing peptides as the authors imply. Apparently, the main reason why the SiO₂@DpaZn bead method is better for N-phosphopeptides is because of the compatibility with alkaline/neutral pH. But in that case, it would not eliminate SiO₂@DpaZn bead binding to other types of phosphopeptides, and therefore the authors need to consider more than N-phosphates.

Reply: Thank you for your kindly comments. We found that SiO₂@DpaZn beads binding other types of phosphopeptides. If we concerned S-phos peptide, the material can be used for S-pho peptide enrichment under optimized condition. We explained the reason why the system prefers N-phos peptide comparing with commonly O-phos peptide in question 1 for reviewer 1. The exact sequences and phosphorylation sites were shown in supporting Data.

Question 3: Page 4, bottom: The convention in writing out peptides containing a phosphoresidue is to use pS, pT, pY, pH, etc. As it stands, by using TSHpYSIMAR, it is unclear whether it is the Tyr or the His residue in the TSHYSIMAR peptide that is phosphorylated! This needs to be changed throughout the paper.

Reply: We do appreciate your kindly suggestion. We apologize for the inappropriate writing of the phosphorylation sites. It was corrected throughout our revised manuscript and supporting information.

Question 4: Page 4: How did the authors rule out the possibility that the Arg in this peptide was also phosphorylated during the chemical phosphorylation reaction with potassium phosphoramidate - MALDI-TOF MS analysis alone would not discriminate between the pHis and pArg forms? In addition, although perhaps less likely, how did they rule out the presence of phosphate on the Ser, Thr and Tyr residues in the chemically phosphorylated peptide? This would require MS/MS analysis to define the b/y ions.

Reply: Thank you for your kindly suggestion. Potassium phosphoramidate cannot react with Arg, Ser, Thr or Tyr residues (Method. Enzymol., 200, 388-414 (1991); Amino Acids, 32, 145-156 (2007); Curr. Protein Pept. Sc., 10, 536-550 (2009)). To figure out the exact phosphorylation site on this peptide, the chemical phosphorylated peptide TGIFpKSAR was introduced to do ESI-MS/MS analysis (Figure 3). The continuous b/y ions around lysine demonstrate the presence of phosphorylation on lysine, but not on the Arg residue. The similar methods can be used for discriminate between the pHis and pArg forms.

Figure 3. ESI-MS/MS spectra of pLys peptide (TGIFpKSAR).

Question 5: Page 4: Were the authors able to measure the relative levels of the phosphopeptide versus the unphosphorylated form of the peptide before enrichment?

Reply: We are appreciated for the reviewer's comment. Due to the low-abundance N-phosphopeptides in biological samples, the direct measurement of N-phosphopeptides are difficult. The relative levels of N-phosphopeptides versus

unphosphorylated forms were still unknown.

Question 6: Figure 2: Since there was no peptide fragmentation in this analysis, the authors should state that they obtained a phosphopeptide with a single phosphate (+80 Da) and made the assumption that it is an N-phosphate solely based on the enrichment conditions, N.B. the neutral/alkaline conditions used here do not cause degradation of pS/pT/pY and so there is no reason to think that N-phosphate peptides were selectively retained. Oslund et al. (ref 35) and Potel et al. (ref 39) used the same peptide sequence as a control, but showed that there was also a pTyr version (debate about whether the ‘triplet’ neutral loss fingerprint could differentiate the pHis from the pTyr).

Reply: We are grateful for the reviewer’s comment. The peptide used in manuscript Fig. 2 was pHis peptides obtained by reaction with potassium phosphoramidate. TS_pHYSIMAR was verified by ESI-MS/MS in our experimental. Furthermore, ESI-MS/MS was carrying out to validate the phosphorylation site of the peptide enriched by our strategy (Figure 4). The continuous b/y ions around lysine demonstrate the presence of phosphorylation on lysine. Also the “Triplet” software was introduced to further validate the pLys site (Figure 5). As we can see from the “Triplet”, three neutral loss peaks (-79.97 Da, -97.98 Da and -115.99) differentiate the pLys from the pTyr. To sum up, ESI-MS/MS experiment and “Triplet” test validated the phosphorylation on lysine residue. This method can also be used for differentiate the pHis from the pTyr.

Figure 4 ESI-MS/MS spectra of pLys peptide (TGIFpKSAR). The zoom-in spectrum demonstrated the precursor ion and its neutral loss ions.

Figure 5 TRIPLET software parameters setting and search results. Top left panel: the MS/MS peak list file is loaded and the neutral loss parameters are entered and searched as literature described (J. Am. Chem. Soc., 136, 12899–12911(2014)). Top right panel: after searching the peaklist file, 36566 MS/MS spectra containing the neutral loss triplet were found. Bottom panel: one of the MS/MS spectra in which the neutral loss fingerprint ions (indicated with red diamonds) are found.

Question 7: Page 6/Figure 2: The speed of enrichment is definitely a major aspect/advantage of the SiO₂@DpaZn bead method, but the justification for this experiment is not obvious. It is not standard to vortex or sonicates cell extracts, even for recovery of pS/pT/pY proteins. Presumably this is intended to solubilize proteins that would otherwise pellet even though extraction was done in 8 M urea. But if the authors really want to compare the impact of vortex and sonication, they should indicate the speed of vortex and the power of sonication (watts/joules).

Reply: We are grateful for your great comment. The vortex or ultrasound mentioned in the manuscript is the inevitable operations in the conventional enrichment procedures like incubation, washing and elution, which would cause damage to the N-phosphopeptides (Fig. 2c in revised manuscript). The vortex was carried out on a vortex mixer MX-S (DragonLab, Shanghai, China) with max speed of 2500 rpm. And ultrasound was accomplished with an ultrasonic water bath (Kunshan Ultrasonic Instruments Co. LTD., Kunshan, China) with power of 100% (300 W).

Question 8: Page 6: For a standard analysis with MaxQuant, the cut-off is usually set at 40. So the level of 20 used here seems very low, knowing that they are generally recalibrated below 70, and 20 seems somewhat arbitrary.

Reply: We do grateful for the reviewer's comment. For database searching via Maxquant, the cutoff score is set as 40. Whereas for Mascot, the cutoff value is usually set as 20 to give a false discovery rate of <1% (Mol. Cell. Proteomics 9, 84–99 (2009); 12, 529-538 (2013); 12, 3339–3349 (2013)).

Question 9: Page 6/Data S1: The authors need to discuss the fact that they also found pSer/pThr/pTyr-containing peptides among the tryptic peptides enriched from the digest of the E. coli lysate, and indicate that this means that the SiO₂@DpaZn beads do not only enrich for peptides with N-phosphates. With regard to the identified pArg and pLys peptides, many of them have a C-terminal pLys or pArg, but since trypsin should not cleave at pArg or pLys it is unclear how these peptides could have been generated or how these assignments could have been made. In terms of the identities of the pHis sites, several appear to be active site pHis residues used for phosphate transfer (e.g. phosphoglucomutase), but did any of them correspond to the active site pHis in two-component His kinases?

Reply: Thank you for your kindly suggestion. O-phosphorylated peptides are also stable under our enrichment condition and can be enriched by our method. The sequences and phosphorylation sites were shown in supporting Data S1. Theoretically, pArg or pLys sites should be not cleaved by trypsin due to the negative and bulked phosphate groups. However, in our study, several credible C-terminal pLys and pArg sites have been identified. To explore the enzymatic cleavage of the pLys peptide, six chemical synthetic pLys peptides were prepared and introduced to tryptic digestion and ESI-MS/MS experiments. We were surprised to find that all six pLys peptides could be cleaved by trypsin to different extent. As shown in Figure 6, after label-free quantification by the MaxQuant software, the trypsin digestion rate of each pLys peptide were calculated, and we found that about 30% of pLys sites could be cleaved by trypsin in each pLys peptide. Because we cannot synthesis pArg peptides *in vitro*, the cleavage experiments were not conducted. We speculated that the same phenomenon was also found for pArg peptides. Among the identified N-phosphorylated proteins previous results, five have kinase activity, and two of which were first discovered.

Figure 6 Trypsin digestion rate of six pLys peptides (MGSTGIGNGIAIPHG_pKLEEDTLR, AGYAEDEVVAVS_pKLGDI EYR, TQLIDVIAE_pKAELSK, AANDDLLNSFWLLDSE_pKGEAR, MITGIQIT_pKAANDDLLNSFWLLDSEKGEAR, FVNILMVDG_pKK) after incubated with trypsin at 37°C for 1 h.

Question 10: The authors could do a better job of comparing the pHis, pLys and pArg-containing *E. coli* protein-derived peptides they identified with those reported in the Oslund et al. and Potel et al. papers.

Reply: We do appreciate your kindly suggestion. In Oslund's and Potel's work, there were 15 and 135 credible pHis sites identified from two special cultured *E. coli* cells (M9 minimal medium, consisting of M9 salts (6 g/L Na₂HPO₄, 3 g/L KH₂PO₄, 0.5 g/L NaCl, 1 g/L NH₄Cl) supplemented with either additional 0.5% (w/v) glucose or 0.5% glycerol), respectively. It should be noticed that in such mediums, phosphorylation sites, including N-phosphorylation sites were highly accumulated (J. Proteome Res., 12, 2611-2621(2013); Nat. Biotechnology, 34, 104-113(2016)). In our work, 11 credible pHis sites from 13 pHis peptides were identified from normal cultured *E. coli* cells (Luria-Bertani medium, consisting of 5 g/L yeast extracts, 10 g/L NaCl and 10 g/L tryptone). Under such condition, it is closer to the real phosphorylation state of *E. coli* in nature. Based on this major difference, we find that 3 pHis sites from 4 pHis peptides and 7 pHis sites from 9 pHis peptides were also identified in Oslund's and Potel's experiment, respectively.

In order to compare the enrichment ability of our method with Oslund's and Potel's, N-phosphopeptides from glucose and glycerol cultured *E.coli* (exponential phase and stationary phase) were enriched by our method. After searching through Maxquant software, setting the score over 40 and score difference over 5, 162 N-phosphopeptides (20 pHis, 67 pLys and 75 pArg) and 101 O-phosphopeptides (39 pSer, 56 pThr and 6 pTyr) with localization probability over 0.75 were identified. Compared with Oslund's and Potel's work, our method shows great complementarity in providing information of pLys and pArg peptides, which were much more unstable under acidic conditions (Anal. Bioanal. Chem., 411, 4159-4166(2019); Sci. China Chem., 62, 708-712(2019)).

There are two reasons to explain the different results between our and Potel's work. On the one hand, in addition to the standard sample pretreatment steps, protein precipitation and benzonase treatment were implied before digestion which significantly reduce the inference of pollutants. On the other hand, pH 2.3 was used in the enrichment processes which increase the affinity ability of Fe-IMAC. These two processes no doubt increase the enrichment ability of pHis. However, considering that pLys and pArg peptides, especially pLys peptides (Anal. Bioanal. Chem., 411, 4159-4166 (2019); Sci. China Chem., 62, 708-712 (2019)) are suffered from severe hydrolysis under long operation and the acidic condition, our method might be more suitable for pLys and pArg peptides enrichment.

Question 11: Page 7: It is not clear whether the BSA peptide with a single phosphate on Ser2 would in fact exhibit a major difference in retention time versus a mono-pHis form of the peptide. They could compare the chemically phosphorylated pHis peptide with a synthetic pSer version of the peptide to establish this. Only a good coverage of the b/y ion series from fragmentation would be able to distinguish the two sites, since the mass and the charge are similar.

Reply: Thanks for your kindly suggestion. We synthesized the pSer peptide (TpSHYSIMAR) as well as pTyr peptide (TSHpYSIMAR), and compared the fragment patterns and retention times of these peptides with the pHis peptide

(TSpHYSIMAR). We could distinguish the phosphorylation sites from their fragment patterns. To be specific, as shown in Figure 7a, the sequential y and z ions around pHis site (especially y6, y7, z6, z7 ions) verified the existence of pHis site. Similarly, the sequential y and z ions around pSer and pTyr sites verified the existence of pSer and pTyr sites (Figure 7b and 7c). We compared the extract ion chromatograms (XICs) of the three phosphopeptides (Figure 8), and found that even though the amino acid sequences of these peptides were identical, different phosphorylation sites would cause distinct retention times (the retention times of pHis, pSer and pTyr peptides were 33.58, 29.71 and 25.03 min, respectively).

Figure 7. MS/MS spectra of three phosphopeptides (TSpHYSIMAR, TpSHYSIMAR, TSHpYSIMAR). The superscript P, N and O demonstrated the neutral loss of phosphate group, NH₃ and H₂O.

Figure 8 XICs of three phosphopeptides (TS_pHYSIMAR, T_pSHYSIMAR, TSH_pYSIMAR). The retention times were labeled and MS spectra were shown.

Question 12: Page 7, bottom: All three of these amino acids can be phosphorylated (two generate non-conventional phosphoamino acids that could potentially be enriched and detected under the neutral/alkaline conditions used here), but apparently the authors did not consider pCys and pAsp in their in silico interrogation.

Reply: We are grateful for the reviewer's comment. Theoretically, pCys, pAsp and pGlu peptides could also be enriched by our method. However, because the elution condition was 1% NH₃·H₂O and pAsp and pGlu peptides were unstable under this alkaline conditions, they could not be identified. Therefore, we re-search the Mascot engine by adding pCys as the variable modification, and 4 pCys peptides with ion score larger than 20 were identified.

Question 13: Page 9: Once again, it is not clear whether SiO₂@DpaZn bead enrichment is specific for N-phosphate, or specific to peptide-linked phosphate in general, working on N-phosphate as well.

Reply: We are grateful for the reviewer's comment. Theoretically, zinc-complex

trap all type phosphorylated peptides including O-phosphorylated peptides. We found that SiO₂@DpaZn can enrich O-phosphorylated peptides and S-phosphorylated peptides. Furthermore, we explained the reason why the system prefers N-phos peptide comparing with other systems in question 1 for reviewer 1.

Question 14: Page 13: Only MALDI-TOF MS was used for synthetic phosphopeptide characterization, and so authors cannot validate the position of the phosphate enriched in their peptides.

Reply: We are grateful for the reviewer's comment. In our experimental, ESI-MS/MS was carrying out to validate the phosphorylation site of the synthetic peptide. In the case of N-phosphorylated peptides was verified by ESI-MS/MS, we used MALDI-TOF MS to quickly characterize the he synthetic peptide.

Question 15: Page 14, line 7: This implies that only MALDI-TOF MS was used to characterize the eluted peptides, but at the bottom of the page it states that NanoRPLC-ESI-MS/MS on a Q-Exactive MS system was used. This needs to be clarified, and a description added for exactly how the existence and position of the phosphate in enriched peptides were established.

Reply: We are grateful for the reviewer's comment. In our experimental, ESI-MS/MS was carrying out to validate the phosphorylation site of the synthetic peptide and E.coil peptides. In the case of N-phosphorylated peptides was verified by ESI-MS/MS, we used MALDI-TOF MS to quickly characterize the he synthetic peptide. So MALDI-TOF MS and NanoRPLC-ESI-MS/MS were simultaneously used in our experiment.

Question 16: Page 14: The precise power (watts/joules) needs to be included.

Reply: We are grateful for the reviewer's comment. The vortex was carried out on a vortex mixer MX-S (DragonLab, Shanghai, China) with max speed of 2500 rpm. And ultrasound was accomplished with an ultrasonic water bath (Kunshan Ultrasonic Instruments Co. LTD., Kunshan, China) with power of 100% (300 W).

Question 17: Page 14: The E. coli protein extracts were denatured at 95°C for 1 min in 8 M urea and PBS (presumably around pH 7, and one would expect this heating to lead to significant loss of phospho-N bond phosphate, which is a concern.

Reply: We are grateful for the reviewer's concern. The pH value of lysate buffer (including 8 M urea and PBS) was 7.4, the nitrogen of N-P bond was not protonated, which hindered the leaving ability of the phosphate group (Ciba Foundation Symposium 57, Elsevier, Amsterdam, pp 117–134(1978)). The high temperature might lead to N-P bond hydrolysis. To elucidate the stability of N-phosphopeptides at this condition, 10 chemical synthetic N-phosphopeptides (sequences in Figure 9) were incubated at 95°C for 1 min. After label-free quantification by the MaxQuant software, we found that the recoveries of 6 N-phosphopeptides were over 97%, and that of the rest 4 peptides were higher than 65% (Figure. 9). The results demonstrated that, for the most of N-phosphopeptides, this short incubation time was inadequate to break the N-P bond. It did not affect N-phosphopeptides identification.

Figure 9 Recoveries of ten N-phosphopeptides (A_pKLES_LVEDLVNR, FAST_pHTDSSAQT_VSLEDYVSR, AGYAEDEVVA_VSpKLG_DIEYR, S_pKATNLLYTR, GGPLADGIVITP_SpHNPPEDGIK, SN_pKPFIYQAPFPMGK, IYAYAFDY_pHEK, TGIF_pKSAR, TC_pHAAIIAR, TS_pHYSIMAR) after incubated in PBS buffer (pH 7.4) at 95°C for 1 min. Error bar demonstrated the RSD value of three paralleled experiments.

Question 18: Page 14: The use of trypsin at a 1:3 ratio with substrate protein is much higher than is typical (1:100), and runs the risk of generating non-tryptic cleavages.

Reply: We are grateful for the reviewer's comment. In a typical sample preparation process (including cell harvest 0.5 h, protein extraction 0.5 h, denature 0.02 h, reduction 1.25 h, alkylation 0.5 h and digestion 10 h), digestion occupies 85% of the whole time. Whereas, N-phosphopeptides suffer from hydrolyze at 37 °C for such a long time. Therefore, more trypsin (1:3 to substrates) was added to minimize the time of digestion without reducing hydrolyze. This "high concentration trypsin" method has been used in phosphoproteome analysis (Anal. Chem., 86, 6786–6791(2014)). In "high concentration trypsin", the digestion time was reduced to 1 h, which significantly reduced the risk of generating non-tryptic cleavages.

Question 19: Page 15, top: The authors need to indicate the pH of the buffers used for the C18 HPLC separation of peptides. Since the buffers contained formic acid they will presumably have been ~ pH 2, which will also lead to loss of phospho-N bond phosphate. In this regard, did the authors identify any nonphosphorylated peptides corresponding to know sites of His phosphorylation, which might represent pHis peptides that were successfully enriched but then lost their phosphate during the HPLC fractionation step.

Reply: Thank you for your kindly suggestion. In the HPLC-ESI-MS/MS analysis on positive mode, 0.1% FA (~ pH 2.0) is usually added in the two mobile phases to ensure the good retention and ionization. After carefully check the list of non-phosphorylated peptides, several of the peptides corresponding to know sites of N-phosphorylation were discovered. It was inevitable in existing separation and identification methods. It should be noticed that silicone separation matrix can protect N-P bonding from hydrolysis at a certain extent by silica adsorption (Anal. Chem, 79, 7450-7456 (2007)). Therefore, during the separation process, since N-phosphorylation peptides remain adsorbed on the matrix for most of the time, only a small percentage of the modifications will be hydrolyzed.

Question 20: Page S12: Twenty is a low cutoff score - generally 40 is a standard; did the authors consider the probability of localization when the scores are different? If the scan number is different, how do they consider the fact that both sites can be phosphorylated on different related phosphopeptides with different scores?

Reply: Thank you for your kindly suggestion. For database searching via Maxquant, the cutoff score is set as 40. Whereas for Mascot, the cutoff value is usually set as 20 (Mol. Cell. Proteomics 9, 84–99(2009); 12, 529-538(2013); 12, 3339–3349(2013)). Since mascot was used for all database search in our previous experiment, 20 was set as the cut-off score. A phosphorylation site localization criterion (Fig S7) was set to confirm the exact sites on the peptides in our previous version, which assumed that there was only one possible phosphorylation site on each mono-phosphopeptide. As you mentioned, if the scan number is different, it is possible that multiple sites may be phosphorylated on a phosphopeptides with different scores. Therefore, in our supplementary experiment of revised manuscript, all the phosphorylation sites with a Maxquant score over 40 and a score difference over 5 were considered phosphorylated, and such screening criteria was widely accepted in phosphorylation proteomics analysis (Nat. Methods, 15, 187–190 (2018); Sci. Rep., 9, 8337-8347 (2019); J. Proteome Res., 7, 5314-5326 (2008)). Therefore, in revised manuscript, N-phosphopeptides from glucose and glycerol cultured *E.coli* (exponential phase and stationary phase) were enriched by our developed method, and 162 N-phosphopeptides (20 pHis, 67 pLys and 75 pArg) and 101 O-phosphopeptides (39 pSer, 56 pThr and 6 pTyr) with localization probability over 0.75 were identified.

Reviewers' comments:

Reviewer #1 (Remarks to the Author):

Responses and the revised manuscript from the author fully satisfied my questions. Therefore, it will meet the requirement for the publication in the journal.

Reviewer #2 (Remarks to the Author):

The authors have satisfyingly addressed my comments and concerns in the revised version of their manuscript. I have no further comments.

Reviewer #3 (Remarks to the Author):

The authors have made a good effort to address the points raised in the reviews, but in my view this new method still needs to be validated using different starting samples, and particularly mammalian cells, to show experimentally that it is as good as or superior to other methods by direct comparison, and determine how selective it is for the N-phosphoproteome versus the O-phosphoproteome.

1. The authors discuss in more depth the issue of whether SiO₂@DpaZn beads also enrich pSer, pThr and pTyr peptides, concluding that it should enrich both N-pho and O-pho peptides, and arguing that the lower number of O-pho peptides they obtained is because *E. coli* lacks protein kinases able to phosphorylate Ser/Thr/Tyr and therefore their phosphorylated forms are underrepresented in the *E. coli* phosphoproteome. However, they did not compare the O-pho sites they identified, with the *E. coli* O-pho sites previously reported in the literature. In order to establish general utility of this new SiO₂@DpaZn method, the authors really need to analyze a tryptic digest of a mammalian cell line lysate, where O-pho sites will greatly predominate over N-pho sites. Moreover, they need to mention early on in the paper, where they discuss the reasons for using SiO₂@DpaZn, that there is no theoretical reason why SiO₂@DpaZn beads should not bind O-pho peptides at neutral pH, otherwise readers may come away with the impression that this method is actually selective for N-pho peptides.

2. The authors did not validate their method by carrying out a comparative experimental analysis using one of the other methods that have been reported for the isolation and MS analysis of N-pho peptides under neutral conditions (e.g. Hardman et al. *EMBO J* 38:e100847, 2019; N.B. this paper was not cited). All they did in this regard was to compare their results with those reported by Potel et al. who identified N-pho peptides from *E. coli* using an accelerated and modified TiO₂ enrichment protocol. However, the authors' comparative analysis (new paragraph on page 8), does not make clear how many of the 135 pHis peptides identified by Patel et al. in *E. coli* were found in their own 162 N-pho peptide dataset, and instead they conclude that their method may be better for pLys and pArg peptides, which were not even analyzed by Potel et al.

3. The pHis/pLys/pArg site identifications appear to be reproducible based on their triplicate analysis and identification by MS/MS using the b/y ions series. However, as noted previously, several of the peptides in Data S1 contain a C-terminal pLys or pArg. Because, as was pointed out in the original reviews, trypsin cleavage at a negatively charged residue seems extremely unlikely, the authors included some data in the rebuttal where they tested the ability of trypsin to cleave synthetic pLys-containing peptides from which they conclude that trypsin can cleave pLys-containing peptides. However, there is no real description of what was actually done in this experiment, e.g. were the chemically phosphorylated peptides demonstrated to be 100% phosphorylated, and can the authors be sure that these peptides did not become dephosphorylated during the digestion, thereby becoming trypsin cleavable; also what ratio of

peptide to trypsin was used. For the analyses described in the paper, they used trypsin at a 1:3 ratio with protein, which is a massively high enzyme to substrate ratio and could lead to nonspecific cleavages. The digestions were analyzed by MS, but it is not clear whether both derivative tryptic peptides were measured - it would need to be the fragment containing pLys. From an enzyme structure and biochemical perspective, it is still very hard to understand how a strongly negatively charged phosphate attached to the ϵ -NH₂ position of Lys would be recognized by the trypsin catalytic hydrophobic pocket which has a negative charged specificity-determining residue at its base (N.B. pLys is also "longer" than Lys, creating another problem for recognition by the enzyme). For these reasons, there are still concerns whether trypsin can cleave at a pLys or pArg bond at any reasonable rate. In fact, in Data S1, the authors show that the majority (7/11) of the pLys peptides they identified had internal pLys residues, indicating that trypsin did not cleave at these sites. Interestingly, two of the pLys peptides they identified with internal pLys, were cleaved in the pLys peptide cleavage assay included in the rebuttal, which is puzzling (also, trypsin is an extremely poor carboxypeptidase, and yet the authors detected apparently efficient cleavage of a peptide ending in pKK). Moreover, the synthetic pLys peptides used for validation shown in Table 1 all have internal pLys residues! A few of the identified pArg peptides also have internal pArg residues, although here the majority appears to have a C-terminal pArg residue. It was for these reasons that Hardman et al. (op. cit.) excluded C-terminal pLys and pArg from their MS data search, although they had identified several in their initial database search.

4. On line 93, the authors mention that they observed a lot of non-phosphopeptides in the MS analysis of SiO₂@DpaZn bead-enriched peptides, but it is unclear whether these peptides represent a reproducible but consistent background, and if so what sort of sequence are enriched, or what fraction of total peptides identified were non-phosphopeptides (as they indicate, a minor fraction of these may be N-pho peptides that became dephosphorylated during the acidic HPLC fractionation step prior to MS analysis).

Other points: 1. There is no specific description of how the synthetic pLys peptides were made, although it appears that this was also done using PPA chemical phosphorylation like the pHis peptides.

2. It is not clear whether SiO₂@DpaZn-beads capture 1-pHis as well as 3-pHis peptides, and the existence of these two distinct pHis isomers is not even mentioned in the paper (!), which has to be rectified. The chemically phosphorylated His-containing synthetic peptides that were generated will contain almost exclusively 3-pHis, because of the long-term incubation with PPA (12 hours). In this regard, was the active site 1-pHis peptide from NDK, the E. coli nucleoside diphosphate kinase, recovered and identified. In their rebuttal, the authors say "Among the identified N-phosphorylated proteins previous results, five have kinase activity, and two of which were first discovered ", but I have no idea what this sentence actually means. Figure S10 depicting the functions of the proteins identified with N-pho sites is not helpful, because it does not segregate pHis, pLys and pArg-containing proteins. From Data S2, it appears that none of the kinases were two-component system (TCS) His kinases, which one would have expected to detect in the E. coli N-phosphoproteome.

3. In the title the authors use the word "comprehensive", but they provide no evidence regarding how comprehensive their analysis is, and, as indicated above, comparison of their data with the Potel et al. data might suggest that their analysis is not comprehensive.

4. The pH/pK/pR nomenclature in the data tables has not been corrected. In Data S2, what was done for the TiO₂ sheet was not described.

Response to the comments of Reviewer 3

Comments

1. The authors discuss in more depth the issue of whether SiO₂@DpaZn beads also enrich pSer, pThr and pTyr peptides, concluding that it should enrich both N-pho and O-pho peptides, and arguing that the lower number of O-pho peptides they obtained is because *E. coli* lacks protein kinases able to phosphorylate Ser/Thr/Tyr and therefore their phosphorylated forms are underrepresented in the *E. coli* phosphoproteome. However, they did not compare the O-pho sites they identified, with the *E. coli* O-pho sites previously reported in the literature. In order to establish general utility of this new SiO₂@DpaZn method, the authors really need to analyze a tryptic digest of a mammalian cell line lysate, where O-pho sites will greatly predominate over N-pho sites. Moreover, they need to mention early on in the paper, where they discuss the reasons for using SiO₂@DpaZn, that there is no theoretical reason why SiO₂@DpaZn beads should not bind O-pho peptides at neutral pH, otherwise readers may come away with the impression that this method is actually selective for N-pho peptides.

Reply: Thank you for your kindly suggestions. Our previous statement was inaccurate. O-phosphorylation may be normally expressed in *E. coli* cell. Lin *et al* reported that 1201 O-pho peptides and 6 pHis peptides were identified from Luria-Bertani medium cultured *E. coli* (Lin, M. H. *etal*, Systematic profiling of the bacterial phosphoproteome reveals bacterium-specific features of phosphorylation *Science Signaling*, **8**, rs10 (2015)). There are totally 12 O-pho sites were identified in our result, among which 8 were also found in Lin's result (Data S1). The limited number of pho peptides is due to the reduced binding force of the SiO₂@DpaZn beads under neutral conditions, compared with acidic enrichment condition. However, as we stated in the last reply, theoretically the material has stronger binding ability to N-phosphorylation under neutral conditions, which can reduce the interference of O-phosphorylation on the identification of N-phosphorylation.

To further validate general utility, SiO₂@DpaZn beads were used to enrich HeLa

cells lysate (Lin et al reported the degree of protein phosphorylation was at least 80 times more than that in bacteria), in which O-pho sites greatly predominate over N-pho sites. Applying the “class I” ptmRS score cut-off 0.75, 4847 N-pho sites (pHis: 1256; pLys: 1854; pArg: 1737, excluding C-terminal pLys and pArg residue) and 7502 O-pho sites (pSer: 3901; pThr: 2523; pTyr: 1078) were identified from tryptic HeLa cells (FDR<1%). The detailed information was listed below the second question.

Furthermore, we mention early in the revised manuscript that SiO₂@DpaZn beads bind both N-pho peptides and O-pho peptides at neutral pH (P3L21-22; P5L1-6), which could make readers have a clearer understanding of enrichment mechanism.

2. The authors did not validate their method by carrying out a comparative experimental analysis using one of the other methods that have been reported for the isolation and MS analysis of N-pho peptides under neutral conditions (e.g. Hardman et al. EMBO J 38:e100847, 2019; N.B. this paper was not cited). All they did in this regard was to compare their results with those reported by Potel et al. who identified N-pho peptides from E. coli using an accelerated and modified TiO₂ enrichment protocol. However, the authors’ comparative analysis (new paragraph on page 8), does not make clear how many of the 135 pHis peptides identified by Patel et al. in E. coli were found in their own 162 N-pho peptide dataset, and instead they conclude that their method may be better for pLys and pArg peptides, which were not even analyzed by Potel et al.

Reply: Thank you for your kindly suggestions. Hardman *et al* reported a strong anion exchange (SAX)-mediated method for revealing human non-canonical phosphorylation. They identified 444 unique N-pho sites from HeLa cell extracts: 134 for pHis; 150 for pLys; and 160 for pArg, compared with 3099, 586 and 147 phosphosites for pSer, pThr and pTyr, respectively (3832; ptmSR \geq 0.75, FDR<1%). It demonstrated the feasibility of neutral enrichment method. Thank you very much for your kind reminder. We cited the excellent work in our revised manuscript (reference

46). In addition, we applied SiO₂@DpaZn beads for mammal cell N-phosphorylation analysis. Totally, 4847 N-pho sites (pHis: 1256; pLys: 1854; pArg: 1737, excluding C-terminal pLys and pArg residue) and 7502 O-pho sites (pSer: 3901; pThr: 2523; pTyr: 1078) were identified from HeLa lysate. Compared with SAX-based strategy, the ratio of O-pho sites to N-pho sites was decreased from 8.63 to 1.55, which demonstrated the good selective for the N-phosphoproteome versus the O-phosphoproteome. Furthermore, motif analysis is conducive to evaluate the features of the N-phosphopeptides in HeLa, and the +/-10 residue sequence windows were generated from all N-pho sites and tested against HeLa proteome background. We observed a notable preference for leucine relative to N-pho sites. Moreover, serine, lysine, glutamic acid were overrepresented around N-pho sites (Fig. 1). We added the above description in revised manuscript (P8L18-34; P9L1-11). Hardman *et al*'s results also verified a preference for leucine around N-pho sites, demonstrating the reliability of two methods. Moreover, there were 11 N-pho sites, 132 O-pho sites and 1151 proteins commonly identified in two methods (ptmSR \geq 0.75). Therefore, the two methods commonly verified the feasibility of neutral enrichment for N-phosphorylation.

There were totally 20 pHis peptides identified from M9 minimal medium cultured *E.coli* by our beads. Compared with Patel *et al*'s results (135 pHis peptides), no overlap is found, which might resulted from the dynamic change of phosphorylation. However, pHis 306 and pHis 265 of pHisfructose-1,6-bisphosphatase I (gene name: fbp) was detected by our and Patel *et al*'s method, respectively, which facilitated pHis sites discovery. Furthermore, there were 7 O-pho sites commonly identified in two methods, demonstrating the complementarity between the two methods. We added the description in revised manuscript (P8L1-5). N-phosphoproteome is an its infancy and the lack of effective enrichment methods hindered the development of N-phosphorylation. Enrichment under neutral condition can significantly inhibit P-N bonding hydrolysis and also contributed for other non-canonical phosphorylation identification.

Fig 1. Motif analysis for pHis, pLys and pArg-containing peptides. The amino acid sequences surrounding confidently sites of (A) pHis, (B) non-C-terminally localised pLys, (B) non-C-terminally localised pArg (ptmRS \geq 0.75) were applied for sequence enrichment using Motif-X. Depicted are the sequences of the enriched motifs.

3. The pHis/pLys/pArg site identifications appear to be reproducible based on their triplicate analysis and identification by MS/MS using the b/y ions series. However, as noted previously, several of the peptides in Data S1 contain a C-terminal pLys or pArg. Because, as was pointed out in the original reviews, trypsin cleavage at a negatively charged residue seems extremely unlikely, the authors included some data in the rebuttal where they tested the ability of trypsin to cleave synthetic pLys-containing peptides from which they conclude that trypsin can cleave pLys-containing peptides. However, there is no real description of what was actually done in this experiment, e.g. were the chemically phosphorylated peptides demonstrated to be 100%

phosphorylated, and can the authors be sure that these peptides did not become dephosphorylated during the digestion, thereby becoming trypsin cleavable; also what ratio of peptide to trypsin was used. For the analyses described in the paper, they used trypsin at a 1:3 ratio with protein, which is a massively high enzyme to substrate ratio and could lead to nonspecific cleavages. The digestions were analyzed by MS, but it is not clear whether both derivative tryptic peptides were measured - it would need to be the fragment containing pLys. From an enzyme structure and biochemical perspective, it is still very hard to understand how a strongly negatively charged phosphate attached to the ϵ -NH₂ position of Lys would be recognized by the trypsin catalytic hydrophobic pocket which has a negative charged specificity-determining residue at its base (N.B. pLys is also “longer” than Lys, creating another problem for recognition by the enzyme). For these reasons, there are still concerns whether trypsin can cleave at a pLys or pArg bond at any reasonable rate. In fact, in Data S1, the authors show that the majority (7/11) of the pLys peptides they identified had internal pLys residues, indicating that trypsin did not cleave at these sites. Interestingly, two of the pLys peptides they identified with internal pLys, were cleaved in the pLys peptide cleavage assay included in the rebuttal, which is puzzling (also, trypsin is an extremely poor carboxypeptidase, and yet the authors detected apparently efficient cleavage of a peptide ending in pKK). Moreover, the synthetic pLys peptides used for validation shown in Table 1 all have internal pLys residues! A few of the identified pArg peptides also have internal pArg residues, although here the majority appears to have a C-terminal pArg residue. It was for these reasons that Hardman et al. (op. cit.) excluded C-terminal pLys and pArg from their MS data search, although they had identified several in their initial database search.

Reply: Thanks for your kindly suggestions. We agreed with the reviewer’s opinion, and excluded C-terminal pLys and pArg from our MS data search in the revised manuscript (P6L10-11; P7L21-22; P8L26). Furthermore, C-terminal pLys and pArg of HeLa cells peptides was assigned as blue background in Data S3.

4. On line 93, the authors mention that they observed a lot of non-phosphopeptides in

the MS analysis of SiO₂@DpaZn bead-enriched peptides, but it is unclear whether these peptides represent a reproducible but consistent background, and if so what sort of sequence are enriched, or what fraction of total peptides identified were non-phosphopeptides (as they indicate, a minor fraction of these may be N-phosphopeptides that became dephosphorylated during the acidic HPLC fractionation step prior to MS analysis).

Reply: Thanks for your kindly suggestions. We deduced that non-phosphopeptides might come from the non-specific adsorption of beads, together with the dephosphorylated peptides prior to MS analysis, which represented reproducible but consistent background, but without special sequence motif, no matter in *E. coli* or HeLa cell lysate.

Other comments

1. There is no specific description of how the synthetic pLys peptides were made, although it appears that this was also done using PPA chemical phosphorylation like the pHis peptides.

Reply: Thanks for your kindly suggestions. According to previous research (Arlen, W. F. *et al.*, Synthesis and properties of N-, O-, and S-phospho derivatives of amino acids, peptides, and protein, *CRC Critical Reviews in Biochemistry*, **16**, 51-101(1984)), PAA can be used to chemical phosphorylation of lysine when the peptide sequence does not contain histidine. Our previous results also verified the chemical phosphorylation of Lys (Hu, Y. C. *et al.* Isolation and identification of phosphorylated lysine peptides by retention time difference combining dimethyl labeling strategy. *Sci China Chem* **62**, 708-712 (2019); Hu, Y. C. *et al.* Cleavable hydrophobic derivatization strategy for enrichment and identification of phosphorylated lysine peptides. *Anal Bioanal Chem* **411**, 4159-4166 (2019)). We added the description in revised supporting experimental procedures.

2. It is not clear whether SiO₂@DpaZn-beads capture 1-pHis as well as 3-pHis peptides, and the existence of these two distinct pHis isomers is not even mentioned

in the paper (!), which has to be rectified. The chemically phosphorylated His-containing synthetic peptides that were generated will contain almost exclusively 3-pHis, because of the long-term incubation with PPA (12 hours). In this regard, was the active site 1-pHis peptide from NDK, the E. coli nucleoside diphosphate kinase, recovered and identified. In their rebuttal, the authors say “Among the identified N-phosphorylated proteins previous results, five have kinase activity, and two of which were first discovered “, but I have no idea what this sentence actually means. Figure S10 depicting the functions of the proteins identified with N-pho sites is not helpful, because it does not segregate pHis, pLys and pArg-containing proteins. From Data S2, it appears that none of the kinases were two-component system (TCS) His kinases, which one would have expected to detect in the E. coli N-phosphoproteome.

Reply: Thanks for your kindly suggestions. In scheme 1, we have showed the chemical structure of two pHis isomers, and we further added the description of the existence of two distinct pHis isomers in our revised manuscript (P2L5; P2L18). According to the previous research (Ojida, A. *et al*, Molecular Recognition and Fluorescence Sensing of Monophosphorylated Peptides in Aqueous Solution by Bis(zinc(II)-dipicolylamine)-Based Artificial Receptors, *J. Am. Chem. Soc.* **126**, 2454-2463 (2004)), the DpaZn(II) has a vacancy on each Zn(II) ions and can access to form RO(N)PO₃-DpaZn(II) complex with the phosphate anion [RO(N)PO₃⁻] under neutral condition. We also found the zeta potential of SiO₂@DpaZn kept stable at about +40 mV around pH 7.0, enabling the additional electrostatic interaction with phosphate groups. The above two binding forces are independent of isomer position. Furthermore, we use PPA to prepare 1-pHis and 3-phs of standard peptide (LIHGQVATR) by a shorter time (1 h). By data search, LI_pHGQVATR(containing 1-pHis and 3-pHis, Table 1) were identified. Therefore, SiO₂@DpaZn can theoretically and experimentally enrich two isomers of pHis.

The sentence “Among the identified N-phosphorylated proteins previous results, five have kinase activity, and two of which were first discovered “ means that there is five identified N-phosphoproteins have kinase activity (in Data S1). For clearer discussion, we deleted this sentence in our revised manuscript. Segregate functions of

pHis, pLys and pArg-containing proteins is lack of significance because of less number of each N-pho proteins. Therefore, we deleted Figure S10. However, we added Motif of N-phosphopeptides from HeLa which is more meaningful and can guide the identification of N-phosphopeptides.

Different culture conditions might affect the phosphorylation status of *E.coli*. This may be result in no two-component system (TCS) His kinases was found in Data S2. Lin *et.al* verified that many O-phosphorylated proteins were also observed among phosphotransferases or TCS that were previously thought to be phosphorylated only on histidine, cysteine, or aspartate (Lin, M. H. *etal*, Systematic profiling of the bacterial phosphoproteome reveals bacterium-specific features of phosphorylation *Science Signaling*, **8**, rs10 (2015)). In our result, we found that sensory histidine kinase in TCS (senses copper ions) with pTyr site was identified in O-phosphopeptides list (Data S2). Furthermore, two-component system His kinase can be identified by Luria-Bertani cultured *E.coli*. Therefore, it was concluded that our method can identify phosphorylated targets in TCS process.

Table 1. Date search of peptide LIHGQVATR by PPA phosphorylation.

Probabilities	Score diff	Score	Amino acid	Phosphorylation Probabilities	Charge
1.0	64.37	111.04	H	LIH(1)GQVATR	2

3. In the title the authors use the word “comprehensive”, but they provide no evidence regarding how comprehensive their analysis is, and, as indicated above, comparison of their data with the Potel et al. data might suggest that their analysis is not comprehensive.

Reply: In the revised manuscript, to avoid the misunderstanding by readers, we deleted “comprehensive” in title in our revised manuscript. The title was change to “Bis(zinc(II)-dipicolylamine) functionalized sub-2 μm core-shell microspheres for analysis of N-phosphoproteome”.

4. The pH/pK/pR nomenclature in the data tables has not been corrected. In Data S2, what was done for the TiO₂ sheet was not described.

Reply: Thank you very much for your kindly suggestions. We carefully corrected the writing format of phosphorylation in data tables. The results of TiO₂ enrichment was listed in Data S1 by mistake, and was deleted in the revised manuscript.

REVIEWER COMMENTS

Reviewer #3 (Remarks to the Author):

The authors have done many of the experiments requested by the reviewer. In particular, they have used their SiO₂@DpaZn bead method to enrich tryptic peptides from a HeLa cell lysate for subsequent MS analysis, which identified a large number of N-pho peptides, although about twice as many O-pho peptides were enriched than N-pho peptides, as one might have expected given the enormous abundance of O-pho sites in mammalian cells. However, the poor concordance between their HeLa cell N-pho dataset and the Hardman et al. HeLa cell N-pho dataset is something of a concern, and one would hardly call having 11 N-pho sites in common out of 460 sites a validation for either method, socially since no well characterized pHis sites were identified in either study. True validation of a subset of these sites is needed, and this will require expressing mutant forms of proteins interest with His/Arg/Lys to Ala mutations to demonstrate that these peptides are no longer enriched, and ideally show that loss of N-phosphorylation at a particular site has a functional consequence, although clearly such studies are beyond the scope of the present paper.

I recommend that the authors address the points below before publication. In addition, rather than discussing the comparisons between their E. coli N-pho database and that of Potel's group, and their HeLa cell N-pho dataset and that of the Evers' group in the text, it would be much easier if they created a single Table listing the peptide numbers from all three studies, and then talk in more general terms about the comparisons in the text. As it stands, putting all the numbers in the text makes it hard to follow the discussion.

1. The authors have now acknowledged that there are two isoforms of pHis, but the experiment they did to try and show that SiO₂@DpaZn beads can bind a 1-pHis control peptide is not interpretable as it stands. To test whether SiO₂@DpaZn beads are able to enrich for 1-pHis peptides, the authors say that they used PPA to prepare 1-pHis and 3-pHis forms of a standard peptide, LIHGQVATR, by using a short incubation time, which is supposed to preferentially generate the 1-pHis isomer. However, without further validation of the LIHGQVATR phosphopeptide population they generated, e.g. using isoform specific pHis antibodies or NMR (N.B., it is not possible to distinguish 1-pHis from 3-pHis using a MS approach), there is no way of knowing for certain whether there was any 1-pHis peptide present in the sample they analyzed. Therefore, this experiment, which is not even described in the paper, is inconclusive. I agree with the authors that there is reason to believe that the SiO₂@DpaZn beads will enrich 1-pHis peptides, but they need to use an authentic 1-pHis peptide to establish this. The easiest way to do this is to use a tryptic digest of autophosphorylated recombinant NME1, and test for SiO₂@DpaZn beads for retention of the 1-pH118 peptide. In this regard, however, the authors did not detect the NME1/2 1-pH118 peptide in their HeLa cell tryptic peptides enriched on SiO₂@DpaZn beads, even though this is an abundant pHis peptide, and this is a good reason for directly validating that the beads can enrich for 1-pHis peptides. Since they have not identified any pHis peptides that have previously been characterized as 1-pHis or 3-pHis (see point 4), they simply ought to state that they do not know whether their for SiO₂@DpaZn bead method enriches both forms of pHis.

2. Page 8 top: It is surprising that there was essentially no overlap between the Potel et al. pHis peptide dataset and theirs. Obviously, the two different enrichment methods could select different subsets of N-pho peptides, but did the authors check whether any unphosphorylated peptides identified in their MS runs on enriched peptides might correspond to N-pho sites that Potel et al. had found to be phosphorylated? Considering that their HPLC separation gradient takes two hours at ~pH 2, it is likely that many enriched N-pho peptides will be dephosphorylated under these extended exposure to acidic conditions. In this connection, although FDPase utilizes a pHis intermediate, neither of the pHis sites they (or Potel et al.) identified in FDPase, pH265 and pH306, is the active site pHis, which is 3-pH13, and this should be stated and discussed. Also, what if anything is known about the possible functions of the pH265 and pH306 residues?

3. Page 8: In their HeLa cell SiO₂@DpaZn enrichment phosphoproteomic analysis the authors found a total of 4847 N-pho sites; 1256 pHis; 1854 pLys; 1737 pArg, which is strikingly large number. They compared this dataset with that of Hardman et al., who using their UPAX method, identified a total of 444 N-pho sites from HeLa cell extracts: 134 pHis; 150 pLys; and 160 pArg. Surprisingly, only 11 N-pho sites were in common between the two data sets. Of course, the use of two different enrichment methods could explain the largely dissimilar N-pho datasets, but if one did this with two different O-pho identification methods one would undoubtedly get a much greater overlap in phosphopeptide identities. There is also a third HeLa cell N-pho dataset available (bioRxiv 2019/691352), which used yet another enrichment method, that could be compared. The one interesting commonality between all three N-pho datasets is that they all exhibited an enrichment for Leu residues surround the N-pho sites. This could either be a preference for the His kinase in question, or alternatively this could be because peptides with Leu residues in the vicinity of the pHis/Arg/Lys residue could be more resistant to hydrolysis.

4. With regard to the HeLa cell N-pho peptide analysis, it is notable that they did not identify the 1-pH118 active site NME1/2 pHis peptide, which is a very abundant pHis peptide in HeLa cells. Similarly, the 1-pH117 active site peptide from the E. coli NDK1 nucleoside diphosphate kinase was missing from their N-pho dataset. In fact, they do not seem to have identified any pHis peptides corresponding to well validated pHis sites, e.g. 3-pHis11 PGAM enzyme intermediate, which is a very abundant pHis enzyme in the glycolytic pathway. In fact, the authors. did not identify the PGAM pHis peptide either in their E. coli or in HeLa N-pho peptide datasets. This the failure to find known pHis sits deserves fuller comment.

Minor point: Supplementary Information, page 2: In the added text, they say "PAA can be used to chemical phosphorylation pLys when the peptide sequence does not have histidine". This should read "PPA can be used to chemically phosphorylate lysine when the peptide sequence does not contain histidine".

Dear reviewer,

Thank you for your kindly suggestions, which are beneficial for improving our manuscript. We made point to point response to your questions.

1. In addition, rather than discussing the comparisons between their *E. coli* N-pho database and that of Potel's group, and their HeLa cell N-pho dataset and that of the Eysers' group in the text, it would be much easier if they created a single Table listing the peptide numbers from all three studies, and then talk in more general terms about the comparisons in the text. As it stands, putting all the numbers in the text makes it hard to follow the discussion.

Reply: According to your kindly suggestion, we made the Table S1 listing the N-pho sites of the *E.coli* and HeLa from the three studies for better comparisons. The detailed description was shown in revised discussion paragraph 2 and 3.

2. The authors have now acknowledged that there are two isoforms of pHis, but the experiment they did to try and show that SiO₂@DpaZn beads can bind a 1-pHis control peptide is not interpretable as it stands. To test whether SiO₂@DpaZn beads are able to enrich for 1-pHis peptides, the authors say that they used PPA to prepare 1-pHis and 3-pHis forms of a standard peptide, LIHGQVATR, by using a short incubation time, which is supposed to preferentially generate the 1-pHis isomer. However, without further validation of the LIHGQVATR phosphopeptide population they generated, e.g. using isoform specific pHis antibodies or NMR (N.B., it is not possible to distinguish 1-pHis from 3-pHis using a MS approach), there is no way of knowing for certain whether there was any 1-pHis peptide present in the sample they analyzed. Therefore, this experiment, which is not even described in the paper, is inconclusive. I agree with the authors that there is reason to believe that the SiO₂@DpaZn beads will enrich 1- pHis peptides, but they need to use an authentic 1-pHis peptide to establish this. The easiest way to do this is to use a tryptic digest of autophosphorylated recombinant NME1, and test for SiO₂@DpaZn beads for retention of the 1-pH118 peptide. In this regard, however, the authors did not detect the NME1/2 1-pH118 peptide in their HeLa cell tryptic peptides enriched on

SiO₂@DpaZn beads, even though this is an abundant pHis peptide, and this is a good reason for directly validating that the beads can enrich for 1-pHis peptides. Since they have not identified any pHis peptides that have previously been characterized as 1-pHis or 3-pHis (see point 4), they simply ought to state that they do not know whether their SiO₂@DpaZn bead method enriches both forms of pHis.

Reply: Thank you for your kindly suggestion. According to 31P NMR monitoring result (Pirrung *et al.*, *J. Org. Chem.* 2000, **65**, 8448-8453), the phosphorylation of histidine itself with PPA proceeds first to give the less stable 1-pHis (kinetic product), which is finally converted to the 3-pHis (thermodynamic product). And the δ difference between 1-pHis and 3-pHis was about 1 ppm. According to your suggestion, we monitored the phosphorylation products of LIHGQVATR by 31P NMR. As shown in Figure 1 (at the end of the text), the product was dominated by 1-pHis within 1 h, and it could be identified by MS after enrichment when peptide was treated with PPA for 20 min (data shown Table 1). Furthermore, as the reviewer stated, the result of biological samples was more convincing than that of standard peptide. We could not find any known 1,3-pHis from the HeLa dataset in our previous manuscript. After reassessing the experiment, we found that the MS spray was not optimal under low ACN gradient. Therefore, we changed ionization device and reanalyzed HeLa lysates. Totally, 3384 N-pho sites (pHis: 611; pLys: 1618; pArg: 1155) and 6635 O-pho sites (pSer: 4105; pThr: 1956; pTyr: 574) were identified from HeLa lysates, among which 2 pHis sites, including MCM3 3-pHis 721 and LMNB1 1-pHis/3-pHis 571, were verified by virtue of both monoclonal pHis antibodies and SAX methods (Fuhs, S. R. *et al.*, *Cell*, 2015, **162**, 198-210; Hardman, G. *et al.*, *EMBO J*, 2019, **38**, e100847). However, we did not identify any known pHis sites including NME1/2 1-pHis 118, PGAM 3-pHis11 and NDK1 1-pH117 from biological samples. Therefore, to be cautious, we stated that we did not know whether our method enriches both forms of pHis from complex biological samples.

Table 1. Data search result of peptide LIpHGQVATR.

Probability	Score diff	Score	pX	Position	Charge
1.0	75.36	93.1	H	LIH(1)GQVATR	3

3. Page 8 top: It is surprising that there was essentially no overlap between the Potel et al. pHis peptide dataset and theirs. Obviously, the two different enrichment methods could select different subsets of N-pho peptides, but did the authors check whether any unphosphorylated peptides identified in their MS runs on enriched peptides might correspond to N-pho sites that Potel et al. had found to be phosphorylated? Considering that their HPLC separation gradient takes two hours at ~pH 2, it is likely that many enriched N-pho peptides will be dephosphorylated under these extended exposure to acidic conditions. In this connection, although FDPase utilizes a pHis intermediate, neither of the pHis sites they (or Potel et al.) identified in FDPase, pH265 and pH306, is the active site pHis, which is 3-pH13, and this should be stated and discussed. Also, what if anything is known about the possible functions of the pH265 and pH306 residues?

Reply: Thank you for your kindly suggestion. There were 27 unphosphorylated peptides in our results which were identified as pHis peptides by Potel *et al.* Hunter *et al.* also proved that several His-containing peptides in their enriched or purified samples have been reported to be phosphorylated (CCT7 H346, GAPDH H111, HIST1H4A H75, LDHA/B H67, NME1/2 H118, PGAM1 H11, RPS3A H232, SUCLG1 H299 and TUBB H105), and their phosphate were lost even in a shorter acidic separation time. Although acidic condition contributes to good separation, it might also increase the risk of hydrolysis of N-pho. To overcome the above problem, photonic crystals based column with ultra-efficiency might be an alternative to dramatically reduce the separation time (Wirth *et al.*, *J. Am. Chem. Soc.* 2012, **134**, 10780-10782). We added the discussion in the revised discussion paragraph 2.

Interestingly, Potel *et al.* and we identified two different pHis sites (H265 and H306) of fructose-1,6-bisphosphatase (FDPase), which was a member of the histidine phosphatase superfamily, with 3-pHis H13 as the active site. According to the previous research (Kuznetsova *et al.*, *J Biol Chem.* 2010, **285**, 21049-21059), His 268 of FDPase YK23 from *saccharomyces cerevisiae* as C-terminal tails lie along the rim of the active site of the same monomer molecule providing additional stabilizing interactions with active site 3-pHis 13 or with the bound substrate. The phosphorylated H265 and H306 at the C-terminal tails of the FDPase from *E.coli* might play the similar role.

4. Page 8: In their HeLa cell SiO₂@DpaZn enrichment phosphoproteomic analysis the authors found a total of 4847 N-pho sites; 1256 pHis; 1854 pLys; 1737 pArg, which is strikingly large number. They compared this dataset with that of Hardman *et al.*, who using their UPAX method, identified a total of 444 N-pho sites from HeLa cell extracts: 134 pHis; 150 pLys; and 160 pArg. Surprisingly, only 11 N-pho sites were in common between the two data sets. Of course, the use of two different enrichment methods could explain the largely dissimilar N-pho datasets, but if one did this with two different O-pho identification methods one would undoubtedly get a much greater overlap in phosphopeptide identities. There is also a third HeLa cell N-pho dataset available (bioRxiv 2019/691352), which used yet another enrichment method, that could be compared. The one interesting commonality between all three N-pho datasets is that they all exhibited an enrichment for Leu residues surround the N-pho sites. This could either be a preference for the His kinase in question, or alternatively this could be because peptides with Leu residues in the vicinity of the pHis/Arg/Lys residue could be more resistant to hydrolysis.

Reply: Thank you for your kindly suggestion. Applying sites localization probability over 0.75, 3384, 781 and 425 N-pho sites were identified by SiO₂@DpaZn, SAX and HAP/pHis mAbs, respectively. As shown in Table 2, the low overlap of N-pho sites was found between the three methods, which might be due to different interactions between phosphorylated peptides and materials under neutral conditions and different

separation and MS conditions. However, these methods provided some complementarities beneficial for new N-pho sites discovery. For example, an interesting result was found about the pArg sites on protein SRRM2. Four pArg sites (R294, R320, R986 and R2103) were found by both SiO₂@DpaZn and SAX. Moreover, 3 (R1494, R2131 and R2396), 6 (R302, R356, R851, R1530, R2119 and R2286) and 1 (R1879) pArg sites were exclusively identified by SAX, SiO₂@DpaZn, and HAP/pHis mAbs, respectively. In addition, the results obtained by the three methods have certain similarities. For instance, all three methods exhibited enrichment of leucine residues surround the N-pho sites (Figure 2a). The possible interpretations were added in our revised manuscript. The similar result was further verified by HEPG2 cell (Figure 2b). Furthermore, the identified N-phosphoproteins by three methods exhibited similar molecular functions, such as ATP binding, ATPase activity, RNA binding, nucleotide binding and protein kinase binding. Therefore, these common properties proved the reliability of the three results and further provided guidance for the follow-up mammalian N-pho research. The detailed description was added in revised result section “Identification of N-phosphorylated Peptides from HeLa Lysates” and revised discussion paragraph 3.

Table 2. Comparisons of the HeLa N-pho sites between the three methods. The commonly identified sites by every two methods were listed. Description: for example, P09651K8, P09651: protein name; K: N-phosphorylated amino acid; 8: phosphorylation position within the protein. There were no sites identified between the three methods.

SAX and HAP/pHis mAbs	SiO ₂ @DpaZn beads and HAP/pHis mAbs	SiO ₂ @DpaZn beads and SAX
P09651K8	Q8IWX7H281 Q9NRM7R624 Q07020K30 O15078K2139 Q9ULV0K773	O15042H69 O43852H39 O43852H49 O60346H583 O60885H1081

	Q00839K251	Q95684H155 P20700H571 P25205H721 P43487H23 P46379H109 Q13573H231 Q96AT1H49 Q8NHH9H32 O43852K59 A8MW92K824 O75152K764 P55010K418 Q12888K1091 Q13428K155 Q6PCB5K241 Q86XP3K105 Q9H1E3K184 Q9NVD7K18 Q9UNE7K22 Q9Y2X3K497 Q9UQ35R294 Q9UQ35R320 Q9UQ35R986 Q9UQ35R2103 Q9H6F5R105 P11274R116 P12694R346 P13861R97 P43243R192 Q93074R632 Q9C0C2R670 Q15637R79 Q03001R7508
--	------------	---

Figure 2. Sequence motif analysis of identified N-pho sites from (a) HeLa and (b) HEPG2 lysates using WebLogo.

5. With regard to the HeLa cell N-pho peptide analysis, it is notable that they did not identify the 1-pH118 active site NME1/2 pHis peptide, which is a very abundant pHis peptide in HeLa cells. Similarly, the 1-pH117 active site peptide from the *E. coli* NDK1 nucleoside diphosphate kinase was missing from their N-pho dataset. In fact, they do not seem to have identified any pHis peptides corresponding to well validated pHis sites, e.g. 3-pHis11 PGAM enzyme intermediate, which is a very abundant pHis enzyme in the glycolytic pathway. In fact, the authors. did not identify the PGAM pHis peptide either in their *E. coli* or in HeLa N-pho peptide datasets. This the failure to find known pHis sits deserves fuller comment.

Reply: Thank you for your kindly suggestion. We carefully checked the dataset of *E. coli* with different conditions and HeLa of our study. There were 3 known pHis sites (pps pHis 421, manX pHis 20, ptsI pHis 189) were successfully identified from Luria-Bertani cultured *E. coli* by our method. However, we missed some known pHis sites, including NME1/2 1-pHis118, NDK1 1-pHis 117 and PGAM 3-pHis11. Actually, Potel *et al.* also did not identify 1-pHis 117 NDK1 and 3-pHis11 PGAM

from M9 minimal medium-cultured *E. coli*, and Eyers *et al.* also did not identify NME1/2 1-pHis118 and PGAM 3-pHis11 from HeLa. Despite successful identification of 3-pHis11 PGAM from HeLa by combining hydroxyapatite and p-His monoclonal antibodies, Hunter *et al.* still failed to identify NME1/2 1-pHis118. We agreed with your comment that the failure to find some known pHis sites deserve in-depth discussion. The possible reasons was added in revised discussion paragraph 4.

Minor point: Supplementary Information, page 2: In the added text, they say “PAA can be used to chemical phosphorylation pLys when the peptide sequence does not have histidine”. This should read “PPA can be used to chemically phosphorylate lysine when the peptide sequence does not contain histidine”.

Reply: Thank you for your kindly suggestion. We revised our description, and we also carefully checked the spelling and language of manuscript.

(a)

(b)

(c)

(d)

Figure 1. ^{31}P NMR spectrum of histidine peptides LIHGQVATR phosphorylation products. (a) PPA, (b) reaction 20 min, (c) reaction 40 min, (d) reaction 4 h and (e)

partial enlarged spectrum at three different time. The new peak 7.60 ppm represents 1-pHis after reaction 20 min. When reaction 4 h, 7.46 and 6.45 represent 1-pHis and 3-pHis, respectively.